# Temporal Continual Learning with Prior Compensation for Human Motion Prediction

**Jianwei Tang**
Sun Yat-sen University
`tangjw7@mail2.sysu.edu.cn`

**Jiangxin Sun**
Sun Yat-sen University
`sunjx5@mail2.sysu.edu.cn`

**Xiaotong Lin**
Sun Yat-sen University
`linxt29@mail2.sysu.edu.cn`

**Lifang Zhang**
Dongguan University of Technology
`2017028@dgut.edu.cn`

**Wei-Shi Zheng**
Sun Yat-sen University
`wszheng@ieee.org`

**Jian-Fang Hu**[*]
Sun Yat-sen University
`hujf5@mail.sysu.edu.cn`

## Abstract

Human Motion Prediction (HMP) aims to predict future poses at different moments according to past motion sequences. Previous approaches have treated the prediction of various moments equally, resulting in two main limitations: the learning of short-term predictions is hindered by the focus on long-term predictions, and the incorporation of prior information from past predictions into subsequent predictions is limited. In this paper, we introduce a novel multi-stage training framework called Temporal Continual Learning (TCL) to address the above challenges. To better preserve prior information, we introduce the Prior Compensation Factor (PCF). We incorporate it into the model training to compensate for the lost prior information. Furthermore, we derive a more reasonable optimization objective through theoretical derivation. It is important to note that our TCL framework can be easily integrated with different HMP backbone models and adapted to various datasets and applications. Extensive experiments on four HMP benchmark datasets demonstrate the effectiveness and flexibility of TCL. The code is available at `https://github.com/hyqlat/TCL`.

## 1 Introduction

Human Motion Prediction (HMP) aims to predict future poses at varied temporal moments based on the observed motion sequences. The accurate prediction of human motion plays a vital role in many applications, such as autonomous driving, human-robot interaction, and security monitoring, enabling the anticipation and mitigation of risks. This task is challenging due to its requirement for predicting multiple moments, including short-term predictions for the "near-future" and long-term predictions for the "far-future".

Previous approaches address this task by autoregressively forecasting using recurrent neural networks (RNNs) and transformer architectures [1, 4, 5, 9–12, 14, 26, 37, 40, 42], or parallelly generating all frames with graph convolution networks (GCNs) [2, 6, 7, 23, 25, 30, 31, 28]. These methods employ

---

[*]Jian-Fang Hu is the corresponding author. Hu is also with Guangdong Province Key Laboratory of Information Security Technology, Guangzhou, China and Key Laboratory of Machine Intelligence and Advanced Computing, Ministry of Education, China.

37th Conference on Neural Information Processing Systems (NeurIPS 2023).

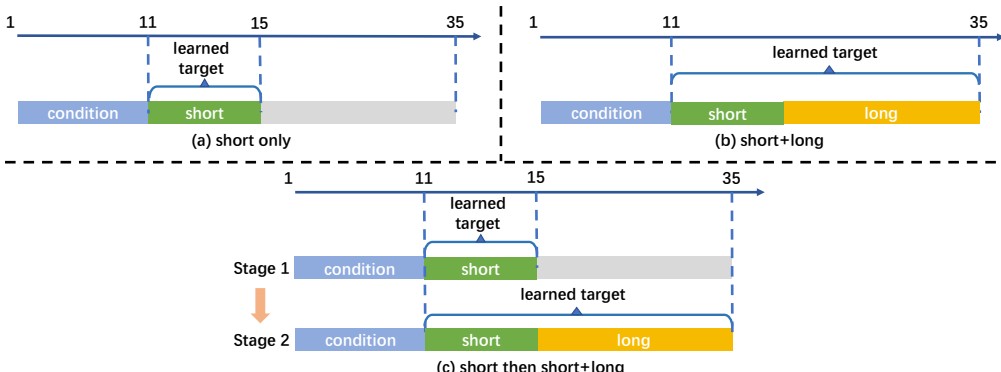

Figure 1: Illustration of three different prediction settings. "short+long" and "short only" represent optimizing prediction for the entire sequence and prediction only for the first 5 frames, respectively. "short then short+long" denotes training the entire sequence after pretraining for the first 5 frames.

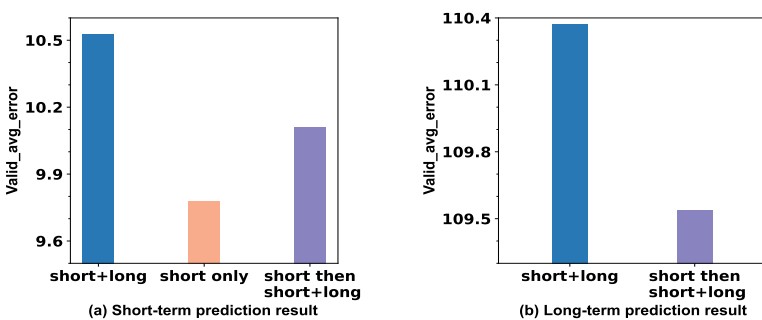

Figure 2: Preliminary experiment results of three different prediction settings (lower values indicate better performance). (a) shows the short-term prediction results (predicting the 2-nd frame), while (b) illustrates the long-term prediction results(predicting the 25-th frame).

the one-stage training strategy to directly train a model that can predict both the short-term prediction and the long-term prediction. However, the long-term motion prediction is more challenging since the future motion can vary greatly (i.e., the prediction space is large), which would increase the uncertainty and ambiguity of future prediction. As the prediction length increases, the fitting of the high-uncertainty long-term prediction will gradually dominate the learning process of the prediction model, which hinders the learning of short-term predictions and further limits the full potential of leveraging the prior knowledge learned from short-term inputs to facilitate long-term predictions.

This motivates us to exploit proper training strategies to better learn and utilize the prior knowledge. We further investigate this by conducting preliminary experiments on the following settings "short+long", "short only", and "short then short+long", which are illustrated in Figure 1. The results are summarized in Figure 2. We observe that "short then short+long" outperforms "short+long" on long-term prediction, which implies that the knowledge learned in short-term prediction can serve as a prior to facilitating the learning of "far-future" prediction. This is intuitive and thus we can use a progressive learning approach, where the model is trained to predict increasing numbers of frames over multiple training stages, e.g., starting with 5 frames in the first stage and 10 frames in the second stage, and so on. However, we also observe that "short then short+long" performs worse than "short only" by a considerable margin for short-term prediction, which demonstrates the joint learning of short-term and long-term prediction results in knowledge forgetting for short-term prediction.

To overcome these problems, we introduce the Prior Compensation Factor (PCF) into the multi-stage training method to obtain a sequential continuous learning framework, which is named Temporal Continual Learning (TCL). It is a multi-stage training framework that alleviates constraint posed by long-term prediction on short-term prediction and effectively utilize prior information from short-term prediction. Specifically, we divide the future sequence into segments and divide the training process

into multiple stages accordingly. We incrementally increase the number of prediction segments, allowing us to leverage the prior knowledge acquired from earlier stages for predicting the subsequent ones. Upon completion of each stage's training, the learned prior knowledge is saved in the model parameters. However, with the changing of optimization objective when switching stages, the prior knowledge gradually fades away. In order to overcome this forgetting problem of prior knowledge, we further introduce the PCF, which is designed as a learnable variable. Then, we derive a more reasonable optimization objective for this regression problem through theoretical derivation.

The proposed training framework is flexible and can be easily integrated with various HMP backbones or adapted to different datasets. To validate the effectiveness and flexibility of our framework, we conduct experiments on four popular HMP benchmark datasets by integrating TCL with several HMP backbones. Our contributions can be summarized as follows:1) We identify certain limitations in existing HMP models and propose a novel multi-stage training strategy called Temporal Continual Learning to obtain a more accurate motion prediction model. 2) We introduce a Prior Compensation Factor to tackle the forgetting problem of prior knowledge, which can be learned jointly with the prediction model parameters. 3) We obtain an easily optimized and more reasonable objective function through theoretical derivation.

## 2    Related Work

**Autoregressive prediction approaches for HMP.**    Motivated by natural language processing, many researchers have adopted sequence-to-sequence models to exploit the temporal information of pose sequences in HMP, which includes RNNs, Long Short-Term Memory Networks (LSTMs) [15] and Transformer [43]. For instance, ERD [8] combined LSTMs with an encoder-decoder to model the temporal aspect, while Jain *et al.*[18] proposed Structural-RNN to capture spatiotemporal features of human motion. Martinez *et al.*[32] applied a sequence-to-sequence architecture for modeling the human motion structure. Aksan *et al.*[1] used Transformer to autoregressively predict future poses. Sun *et al.*[40] designed a query-read process to retrieve some motion dynamics from the memory bank. Lucas *et al.*[27] proposed a GPT-like[35] autoregressive method to generate human poses. Tang *et al.*[42] combined attention mechanism and LSTM to complete the human motion prediction task. However, autoregressive methods are difficult to train and suffer from error accumulation problem.

**Parallel prediction approaches for HMP.**    Some researchers employed parallel prediction methods to address HMP problem [2, 6, 7, 23, 25, 30, 31, 28]. The works of [21–23] used GCN to encode feature or to decode it, which associates different joints' information. Mao *et al.*[31] viewed a pose as a fully connected graph and used GCN to extract hidden information between any pair of joints. Martinez *et al.*[33] devised a transformer-based network to predict human poses. Sofianos *et al.*[39] proposed a method to extract spatiotemporal features using GCNs. And Ma *et al.*[28] tried to achieve better prediction results using a progressive manner. Xu *et al.*[45] used multi-level spatial-temporal anchors to make diverse predictions.

**Continual learning.**    Although Deep Neural Networks (DNNs) have demonstrated impressive performance on specific tasks, their limitations in handling diverse tasks hinder their broader applicability. Therefore, some researchers introduced the concept of Continual Learning (CL)[36] to DNNs to ensure that models retain the knowledge of previous tasks while learning new tasks. Kirkpatrick *et al.*[20] proposed the Elastic Weight Consolidation (EWC) method to overcome the catastrophic forgetting problem and improve the performance of multi-task problems. Shin *et al.*[38] introduced a method that addresses catastrophic forgetting in sequential learning scenarios by using a generative model to replay data from past tasks during the training of new tasks. It is important to note that the traditional CL approaches do not account for temporal correlation and are unable to leverage data from previous tasks.

## 3    Method

The problem of human motion prediction involves predicting future motion sequences by utilizing previously observed motion sequences. Formally, let $\mathbf{X}_{1:T_h} = [\mathbf{X}_1, \mathbf{X}_2, \cdots, \mathbf{X}_{T_h}] \in \mathbb{R}^{J \times D \times T_h}$ denotes the observed motion sequence of length $T_h$ where $\mathbf{X}_i$ indicates motion of time $i$, and $\mathbf{X}_{T_h+1:T_h+T_p} = [\mathbf{X}_{T_h+1}, \mathbf{X}_{T_h+2}, \cdots, \mathbf{X}_{T_h+T_p}] \in \mathbb{R}^{J \times D \times T_p}$ represents the motion sequence of

length $T_p$ that needs to be predicted. Note that $J$ is the number of joints for each pose, and $D$ is the dimension of coordinates. It can be regarded as a composite task consisting of multiple sequential prediction tasks, which involves predicting the future poses at varied moments conditioned on the motion sequences observed in the past.

To accomplish this multiple sequential prediction task, we first model the HMP problem as solving the following optimization problem:

$$\boldsymbol{\theta}^* = \arg\max_{\boldsymbol{\theta}} P(\mathbf{X}_{T_h+1}, \mathbf{X}_{T_h+2}, \cdots, \mathbf{X}_{T_h+T_p} | \mathbf{X}_1, \mathbf{X}_2, \cdots, \mathbf{X}_{T_h}; \boldsymbol{\theta}). \tag{1}$$

Thus our target is to find the optimal model that maximizes Equation (1). To achieve this, we propose a framework called Temporal Continual Learning. Specifically, we partition the entire prediction interval into several smaller segments and perform multi-stage training. We claim that this enables the utilization of prior information from previous segments as knowledge for predicting the subsequent segments. Further, as the optimization objective changes in each training stage, we find that the prior knowledge, i.e., information learned in previous training stages, will be forgotten to a certain degree. To mitigate this problem, we introduce a Prior Compensation Factor and accordingly derive a more reasonable optimization objective at each stage.

### 3.1 Multi-stage Training Process.

We initially decouple the future sequence into $K$ segments with time boundaries $T_{Z_1}, T_{Z_2}, \cdots, T_{Z_K}$, where $T_{Z_K} = T_h + T_p$. And we denote the prediction of segment $k$ as task $Z_k$, which can be expressed as follows:

- Task $Z_1$ :    $\mathbf{X}_{1:T_h} \rightarrow \mathbf{X}_{T_h+1:T_{Z_1}}$
- Task $Z_2$ :    $\mathbf{X}_{1:T_h} \rightarrow \mathbf{X}_{T_{Z_1}+1:T_{Z_2}}$
  $\cdots$
- Task $Z_K$ :    $\mathbf{X}_{1:T_h} \rightarrow \mathbf{X}_{T_{Z_{K-1}}+1:T_{Z_K}}$

To be specific, the target of task $Z_1$ is to predict $\mathbf{X}_{T_h+1:T_{Z_1}}$ conditioned on $\mathbf{X}_{1:T_h}$, and task $Z_k$ aims to predict $\mathbf{X}_{T_{Z_{k-1}}+1:T_{Z_k}}$ with $\mathbf{X}_{1:T_h}$ as condition. Therefore, by leveraging bayesian formulation, optimization problem $P(\mathbf{X}_{T_h+1}, \mathbf{X}_{T_h+2}, \cdots, \mathbf{X}_{T_h+T_p} | \mathbf{X}_1, \mathbf{X}_2, \cdots, \mathbf{X}_{T_h}; \boldsymbol{\theta})$ can be formulated as:

$$P(Z_1 Z_2 \cdots Z_K; \boldsymbol{\theta}) = P(Z_K | Z_1 Z_2 \cdots Z_{K-1}; \boldsymbol{\theta}) P(Z_{K-1} | Z_1 Z_2 \cdots Z_{K-2}; \boldsymbol{\theta}) \cdots P(Z_1; \boldsymbol{\theta}), \tag{2}$$

where $\boldsymbol{\theta}$ is model parameters to be learned. In the following, we denote "$Z_1 Z_2 \cdots Z_k$" as "$Z_{1:k}$". Our target is to maximize Equation (2) which means finding optimal model to accomplish all tasks.

For the purpose of transferring the prior knowledge in preceding tasks to their subsequent prediction task, we progressively increase the number of tasks in temporal order and train them successively. More precisely, our training is decomposed into $K$ stages. In each stage $S_k$, we leverage the optimal model parameters $\boldsymbol{\theta}_{k-1}^*$ trained in the previous stage $S_{k-1}$ to initialize the parameters $\boldsymbol{\theta}$, and then update it based on the prediction tasks $Z_1, Z_2, \cdots, Z_k$ (maximizing $P(Z_{1:k}; \boldsymbol{\theta})$). Since the model is trained to optimize the prediction tasks $Z_{1:k-1}$ in training stage $S_{k-1}$, the knowledge of tasks $Z_{1:k-1}$ can be implicitly involved in its well-trained parameters $\boldsymbol{\theta}_{k-1}^*$. Initializing $\boldsymbol{\theta}_k$ as $\boldsymbol{\theta}_{k-1}^*$ can exploit the prior knowledge learned in previous tasks to assist the prediction of the next task.

### 3.2 Definition of Prior Compensation Factor.

With training different optimization objectives stage by stage, the prior knowledge provided by previous tasks can be effectively exploited to predict the subsequent task. However, the change of the optimization objective in different training stages could also bring about the knowledge forgetting problem. To mitigate the problem, we introduce $\alpha_{Z_{1:k-1} \rightarrow Z_k}$ to estimate the extent of forgotten knowledge when utilizing prior knowledge from tasks $Z_{1:k-1}$ to predict task $Z_k$, which we refer to as the "Prior Compensation Factor".

$$\alpha_{Z_{1:k-1} \rightarrow Z_k} = P(Z_k | Z_{1:k-1}; \boldsymbol{\theta}) - P(Z_k | \hat{Z}_{1:k-1}; \boldsymbol{\theta}). \tag{3}$$

Here, $\hat{Z}_{1:k-1}$ is regarded as the prior knowledge that is reserved and can be still provided for predicting task $Z_k$. So $\hat{Z}_{1:k-1}$ initially represents the prior knowledge reserved in $\boldsymbol{\theta}_{k-1}^*$ in every stage

$S_k$ and would get somewhat corrupted gradually during training. $P(Z_k|Z_{1:k-1};\boldsymbol{\theta})$ indicates the most ideal case, where the current prediction task $Z_k$ can fully leverage the prior information provided by previous prediction tasks $Z_{1:k-1}$. Consequently, the loss of the prior knowledge is non-negative, implying that $0 \leq \alpha_{Z_{1:k-1} \to Z_k} \leq 1 - P(Z_k|\hat{Z}_{1:k-1};\boldsymbol{\theta})$. Specifically, we can observe that $\alpha = 0$ when $P(Z_k|Z_{1:k-1};\boldsymbol{\theta}) = P(Z_k|\hat{Z}_{1:k-1};\boldsymbol{\theta})$, which implies that all the prior knowledge of previous tasks is completely exploited although the $\boldsymbol{\theta}$ changes. By substituting Equation (3) into Equation (2) and taking the negative logarithm, we can obtain:

$$-\log P(Z_{1:k};\boldsymbol{\theta}) = -\log P(Z_1;\boldsymbol{\theta}) - \sum_{i=2}^{k} \log(P(Z_i|\hat{Z}_{1:i-1};\boldsymbol{\theta}) + \alpha_{Z_{1:i-1} \to Z_i}). \tag{4}$$

Our objective is to minimize $-\log P(Z_{1:k};\boldsymbol{\theta})$ with respect to the model parameter $\boldsymbol{\theta}$ and prior compensation factors $\{\alpha_{Z_{1:i-1} \to Z_i}, i = 2, 3, \cdots, k\}$.

## 3.3 Optimization Objective

Optimizing Equation (4) directly is challenging due to the presence of the PCF $\alpha$ and $P(Z_k|\hat{Z}_{1:k-1};\boldsymbol{\theta})$ inside the logarithm. By applying Lemma 3.1 (details provided in the appendix), we can obtain an upper bound for Equation (4), which can be expressed as:

$$\begin{aligned} UB = -\log P(Z_1;\boldsymbol{\theta}) + \sum_{i=2}^{k}((1 - \alpha_{Z_{1:i-1} \to Z_i})(-\log P(Z_i|\hat{Z}_{1:i-1};\boldsymbol{\theta})) \\ + (1 - \alpha_{Z_{1:i-1} \to Z_i})\log(1 - \alpha_{Z_{1:i-1} \to Z_i}) + \log(1 + \alpha_{Z_{1:i-1} \to Z_i})). \end{aligned} \tag{5}$$

Hence, we can turn to minimize the upper bound of Equation (4). It appears that in the optimization objective, $\alpha_{Z_{1:i-1} \to Z_i}$ serves as a factor to control the weights of different tasks, which mitigates the loss of prior information and thus compensates for the lost prior knowledge. The Lemma 3.2 indicates that the largest difference between $-\log P(Z_{1:k};\boldsymbol{\theta})$ and the upper bound $UB$ would not exceed $\log(3/2) * (k-1)$ when $P(Z_i|\hat{Z}_{1:i-1};\boldsymbol{\theta}) \geq 1/2, i \in \{2, 3, \cdots, k\}$.

**Lemma 3.1.** For $0 \leq a \leq 1 - b$ and $0 < b \leq 1$, the inequality $-\log(a + b) \leq (1-a)(-\log b) + (1-a)\log(1-a) + \log(1+a)$ holds. The equality holds if and only if $a = 0$.

**Lemma 3.2.** The absolute difference between the target objective (Equation (4)) and the upper bound (Equation (5)) is not larger than $\log(3/2) * (k-1)$ when $P(Z_i|\hat{Z}_{1:i-1};\boldsymbol{\theta}) \geq 1/2, i \in \{2, 3, \cdots, k\}$. This bound is achieved when $P(Z_k|\hat{Z}_{1:k-1};\boldsymbol{\theta}) = 1/2$ and $\alpha_{Z_{1:i-1} \to Z_i} = 1/2, i \in \{2, 3, \cdots, k\}$.

Due to the space limitation, we present the proofs of Lemma 3.1 and 3.2 in the appendix.

**An intuitive explanation.** Figure 3 illustrates the comparison between the term $-\log(P(Z_k|\hat{Z}_{1:k-1};\boldsymbol{\theta}) + \alpha_{Z_{1:k-1} \to Z_k})$ in the actual optimization objective, the term $-\log P(Z_k|\hat{Z}_{1:k-1};\boldsymbol{\theta})$ in the naive optimization objective merely leveraging multi-stage training strategy, and the corresponding approximate term in our optimization objective. It can be observed that our approximation method has a smaller difference from the actual optimization objective compared to the naive method. This is important as a more reasonable objective function can improve the accuracy of the optimization process. When the prior compensation factor $\alpha_{Z_{1:k-1} \to Z_k}$ is zero, which means $P(Z_k|Z_{1:k-1};\boldsymbol{\theta}) = P(Z_k|\hat{Z}_{1:k-1};\boldsymbol{\theta})$, the prior prediction information from previous tasks $Z_{1:k-1}$ is not lost. As the value of $\alpha$ increases, the discrepancy between the actual objective $P(Z_k|Z_{1:k-1};\boldsymbol{\theta})$ and the objective of the naive method $P(Z_k|\hat{Z}_{1:k-1};\boldsymbol{\theta})$ becomes more evident, indicating a greater degree of forgetting prior knowledge. In contrast, our approach narrows this gap by effectively mitigating the loss of prior information obtained from tasks $Z_{1:k-1}$.

## 3.4 Optimization Strategy

We train the model in a multi-stage manner, in which an initial stage and $K - 1$ TCL stages are involved. The initial stage $S_1$ aims to forecast the motion in the foremost segment, while each TCL stage $S_k, k \in \{2, 3, \cdots, K\}$ performs prediction for segments $Z_{1:k}$ and simultaneously trains

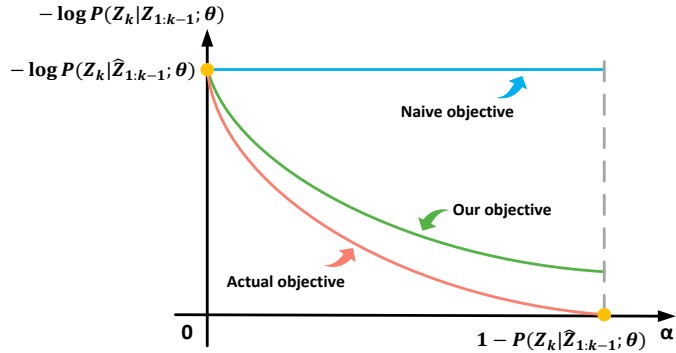

Figure 3: A toy example illustrating the optimization objectives. Our approximate optimization objective is closer to the actual objective compared to the naive optimization method.

$\alpha_{Z_{1:k-1}\to Z_k}$. Once the stage $S_k$ is completely trained, we then estimate the factors $\hat{\alpha}_{Z_{1:k-1}\to Z_k}$ which would be used in the optimization of the following stages. This process is repeated until the final stage $S_K$ is reached.

**Learning of initial stage $S_1$.** Following the implementations of previous methods [19], we can train the initial stage $Z_1$ with Mean Suqared Error (MSE) loss:

$$\mathcal{L}_1 = \sum_{i=T_h+1}^{T_{Z_1}} \left\| \mathbf{X}_i - \hat{\mathbf{X}}_i \right\|^2 \tag{6}$$

where $\mathbf{X}_i$ and $\hat{\mathbf{X}}_i$ represent the ground truth and predicted motion of the $i$-th frames respectively.

**Temporal Continual Learning at stage $S_k$.** In stage $S_k$ ($k \geq 2$), we need to update the model parameters $\boldsymbol{\theta}$ corresponding to tasks $Z_{1:k}$ and the PCF $\alpha_{Z_{1:k-1}\to Z_k}$. According to Equation (5), the loss function in this stage can be calculated as follows:

$$\begin{aligned}
\mathcal{L}_k =& (1 - \alpha_{Z_{1:k-1}\to Z_k}) \sum_{i=T_{Z_{k-1}}+1}^{T_{Z_k}} \left\| \mathbf{X}_i - \hat{\mathbf{X}}_i \right\|^2 + (1 - \alpha_{Z_{1:k-1}\to Z_k}) \log(1 - \alpha_{Z_{1:k-1}\to Z_k}) \\
&+ \log(1 + \alpha_{Z_{1:k-1}\to Z_k}) + \sum_{j=2}^{k-1} (1 - \hat{\alpha}_{Z_{1:j-1}\to Z_j}) \sum_{i=T_{Z_{j-1}}+1}^{T_{Z_j}} \left\| \mathbf{X}_i - \hat{\mathbf{X}}_i \right\|^2 + \mathcal{L}_1
\end{aligned} \tag{7}$$

where the parameters $\hat{\alpha}_{Z_1\to Z_2}, \cdots, \hat{\alpha}_{Z_{1:k-2}\to Z_{k-1}}$ are dertermined in the learning of previous stages. Once the model parameters for stage $S_k$ are determined, we then calculate $\hat{\alpha}_{Z_{1:k-1}\to Z_k}$ as:

$$\hat{\alpha}_{Z_{1:k-1}\to Z_k} = \frac{1}{M} \sum_{m=1}^{M} \hat{\alpha}_{Z_{1:k-1}\to Z_k}^m \tag{8}$$

where $M$ represents number of samples and $\hat{\alpha}_{Z_{1:k-1}\to Z_k}^m$ is PCF estimated for the $m$-th sample.

We continue the TCL training process by predicting stage $S_{k+1}$ and updating the model parameter $\boldsymbol{\theta}$ as well as PCF $\alpha_{Z_{1:k}\to Z_{k+1}}$. We repeat this TCL process until we reach the final stage $S_K$. In practice, we require the backbone model to output an extra dimension and pass it through an MLP head to obtain $\alpha$. The algorithm flow is summarized in Algorithm 1.

**Algorithm 1** Training procedure of proposed TCL framework.

---

**Require:** observed frames $\mathbf{X}_{1:T_h}$, ground truth future frames $\mathbf{X}_{T_h+1:T_h+T_p}$, model parameters $\boldsymbol{\theta}$, stage number $K$, training epoch for $k$-th stage $E_k$, learning rate $\lambda$, training sample number $M$.

    **for** $i = 1$ to $E_1$ **do**

        $\hat{\mathbf{X}}_{T_h+1:T_{Z_1}} = \mathbf{f}_{\boldsymbol{\theta}}(\mathbf{X}_{1:T_h})$

        $\boldsymbol{\theta} \leftarrow \boldsymbol{\theta} - \lambda * \nabla_{\boldsymbol{\theta}} \mathcal{L}_1(\mathbf{X}_{T_h+1:T_{Z_1}}, \hat{\mathbf{X}}_{T_h+1:T_{Z_1}})$

    **end for**

    $\mathbf{A} = \emptyset$

    **for** $k = 2$ to $K$ **do**

        **for** $i = 1$ to $E_k$ **do**

            $\hat{\mathbf{X}}_{T_h+1:T_{Z_k}}, \alpha_{Z_{1:k-1} \to Z_k} = \mathbf{f}_{\boldsymbol{\theta}}(\mathbf{X}_{1:T_h})$

            $\boldsymbol{\theta} \leftarrow \boldsymbol{\theta} - \lambda * \nabla_{\boldsymbol{\theta}} \mathcal{L}_k(\mathbf{X}_{T_h+1:T_{Z_k}}, \hat{\mathbf{X}}_{T_h+1:T_{Z_k}}, \alpha_{Z_{1:k-1} \to Z_k}, \mathbf{A})$

        **end for**

        $\hat{\alpha}_{Z_{1:k-1} \to Z_k} = 0$

        **for** $m = 1$ to $M$ **do**

            $\alpha^m_{Z_{1:k-1} \to Z_k} = \mathbf{f}_{\boldsymbol{\theta}}(\mathbf{X}^m_{1:T_h})$

            $\hat{\alpha}_{Z_{1:k-1} \to Z_k} = \hat{\alpha}_{Z_{1:k-1} \to Z_k} + \alpha^m_{Z_{1:k-1} \to Z_k}$

        **end for**

        $\hat{\alpha}_{Z_{1:k-1} \to Z_k} = \frac{1}{M} \hat{\alpha}_{Z_{1:k-1} \to Z_k}$

        $\mathbf{A} = \mathbf{A} \cup \{\hat{\alpha}_{Z_{1:k-1} \to Z_k}\}$

    **end for**

---

# 4 Experiments

## 4.1 Experimental Setup

We validate our framework on four benchmark datasets. Human3.6M[17] is a large dataset that contains 3.6 million 3D human pose data. 15 types of actions performed by 7 actors(S1, S5, S6, S7, S8, S9 and S11) are included in this dataset. Each actor is represented by a skeleton of 32 joints. However, following the data preprocessing method proposed in [28, 31], we only use 22 joints. The global rotations and translations of poses are removed, and the frame rate is downsampled from 50 fps to 25 fps. For testing and validation, we use actors S5 and S11, while training is conducted on the remaining sections of the dataset. CMU-MoCap is a smaller dataset that has 8 different action categories. The global rotations and translations of the poses are also removed. Each pose contains 38 joints, but following the data preprocessing methods in [28, 31], we only use 25 joints. 3DPW[44] is a challenging dataset that contains human motion data captured from both indoor and outdoor scenes. Poses in this dataset are represented in 3D space, with each pose containing 26 joints. However, only 23 of these joints are used, as the other three are redundant. The Archive of Motion Capture as Surface Shapes (AMASS)[29] dataset gathers 18 existing mocap datasets. Following [39], we select 13 from those and take 8 for training, 4 for validation and 1 (BMLrub) as the test set. We consider forecasting the body joints only and discard those 4 static ones, leading to an 18-joint human pose.

Following the benchmark protocols, we use the Mean Per Joint Position Error (MPJPE) in millimeters (ms) as our evaluation metric for 3D coordinate errors and Euler Angle Error (EAE) for Euler angle representations. The performance is better if this metric is smaller.

**Implementation Details.** Following [28, 30, 39], we set the input length to 10 frames and the predictive output to 25 frames for Human3.6M, AMASS and CMU-Mocap datasets, respectively. For the 3DPW dataset, we predict 30 frames conditioned on the observation of the preceding 10 frames. We choose PGBIG as our backbone model by default. In order to learn the PCF, we add an extra dimension to the output of the backbone model and calculate PCF through an MLP network whose hidden dimension is set to 512. We partitioned the future sequences into three segments with lengths of 3, 9, and 13. The training process was conducted on an NVIDIA RTX 3090 GPU for 120 epochs, allocating 50, 90, and 120 epochs for each respective stage.

Table 1: Results on Human3.6M. *Using EAE as the metric.

| dataset | H36M | | | | | |
|---|---|---|---|---|---|---|
| ms | 80 | 160 | 320 | 400 | 560 | 1000 |
| POTR* | 0.235 | 0.581 | 0.990 | 1.143 | 1.362 | 1.826 |
| POTR+Ours | 0.233 | 0.549 | 0.923 | 1.056 | 1.290 | 1.746 |
| R.S. | 29.9 | 55.7 | 94.5 | 108.7 | 130.1 | 168.0 |
| R.S.+Ours | 28.0 | 52.8 | 91.3 | 105.6 | 127.3 | 166.3 |
| LTD | 12.7 | 26.1 | 52.3 | 63.5 | 81.6 | 114.3 |
| LTD+Ours | 10.8 | 23.0 | 48.2 | 59.3 | 77.5 | 111.2 |
| MM | 12.7 | 26.4 | 53.4 | 65.0 | 83.6 | 117.6 |
| MM+Ours | 11.5 | 25.0 | 52.0 | 63.7 | 82.8 | 117.1 |
| siMLPe | 10.7 | 23.9 | 50.7 | 62.6 | 82.0 | 116.0 |
| siMLPe+Ours | 10.0 | 22.9 | 49.4 | 61.2 | 80.6 | 114.6 |
| PGBIG | 10.3 | 22.7 | 47.4 | 58.5 | 76.9 | 110.3 |
| PGBIG+Ours | **9.4** | **21.3** | **45.7** | **56.8** | **75.4** | **108.8** |

Table 2: Results on CMU-MoCap, 3DPW and AMASS.

| dataset | CMU | | | | |
|---|---|---|---|---|---|
| ms | 80 | 320 | 400 | 560 | 1000 |
| R.S. | 24.4 | 80.1 | 93.1 | 112.5 | 141.5 |
| R.S.+Ours | 21.4 | 72.7 | 85.6 | 105.6 | 136.1 |
| LTD | 9.3 | 33.0 | 40.9 | 55.8 | 86.2 |
| LTD+Ours | 9.1 | 31.0 | 38.5 | 53.1 | 84.9 |
| PGBIG | 7.6 | 29.0 | 36.6 | 50.9 | 80.1 |
| PGBIG+Ours | **7.5** | **28.3** | **35.4** | **48.6** | **78.4** |
| dataset | 3DPW | | | | |
| ms | 200 | 400 | 600 | 800 | 1000 |
| R.S | 99.3 | 129.7 | 142.9 | 161.1 | 171.7 |
| R.S+Ours | 91.8 | 120.0 | 139.5 | 156.2 | 169.1 |
| LTD | 35.6 | 67.8 | 90.6 | 106.9 | 117.8 |
| LTD+Ours | 33.1 | 64.1 | 86.1 | 100.4 | 110.0 |
| PGBIG | 29.3 | 58.3 | 79.8 | 94.4 | 104.1 |
| PGBIG+Ours | **21.4** | **47.1** | **67.7** | **83.5** | **96.0** |
| dataset | AMASS | | | | |
| ms | 80 | 320 | 400 | 560 | 1000 |
| STSGCN | 11.4 | 37.2 | 43.8 | 53.8 | 69.7 |
| STSGCN+Ours | **10.9** | **36.7** | **42.8** | **52.4** | **68.1** |

## 4.2 Experimental Results

We apply our method on the following approaches on four benchmark datasets: Res.Sup.(R.S.) [32], LTD [31], POTR [33], STSGCN [39], MotionMixer(MM) [3], siMLPe [13] and the current state-of-the-art PGBIG [28]. Res.Sup. is an RNN-based model. LTD, STSGCN and PGBIG are GCN-based models. POTR is a transformer-based method. MotionMixer and siMLPe are MLP-based models. All these methods have released their code publicly. We employ their pre-trained models or re-train their models using the suggested hyper-parameters for a fair comparison. And we also follow the metric they used to evaluate the results.

Table 1 presents experimental results of different backbone models before and after applying our training strategy on Human3.6M dataset. Our frameworks outperform the corresponding backbone models by a considerable margin (more than 1.5). Specifically, when applying our strategy to PGBIG, we obtained a performance of 64.97, which is much better than the performance of the original PGBIG (66.52). Table 2 shows the quantitative comparisons of prediction results on the CMU-MoCap, 3DPW and AMASS datasets in which our proposed framework also achieves the best results. We can observe that the improvement in long-term prediction is greater compared to short-term prediction, indicating that the prior information provided by short-term prediction is more crucial for challenging long-term prediction tasks. It is worth noticing that our framework outperforms the others by a significant margin on 3DPW dataset. Specifically, our strategy outperforms PGBIG by 8.9 (63.59 vs 72.49). We attribute this to our framework's ability to leverage prediction prior for the subsequent prediction in this challenging dataset.

## 4.3 Visualization

**Predicting results.** Figure 4 provides some visualization examples of predicted motions, demonstrating that our framework achieves more accurate results. Specifically, Figure 4a represents the action "directions", with the person maintaining an upright position throughout the sequence. The results of PGBIG exhibit a bent posture during long-term predictions, whereas our method can predict states closer to the ground truth position. While in Figure 4b, the person first bends and then stands upright. The PGBIG maintains the bent posture throughout the long-term prediction and our method can accurately predict the changes in posture. It is evident that our method demonstrates an improvement in long-term prediction effectiveness.

$\alpha$ **at different stages.** As Figure 5 displays, the value of $\alpha$ progressively increases with each training stage. When a new task is added, the model focuses on the new objective without considering

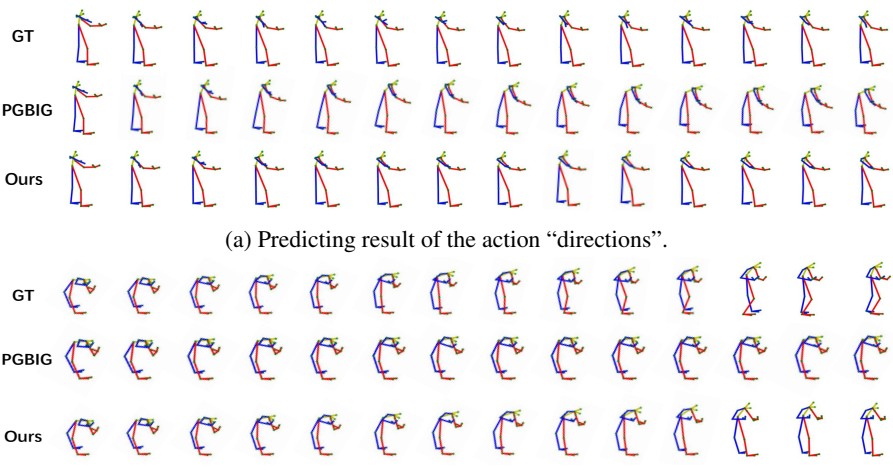

(a) Predicting result of the action "directions".

(b) Predicting result of the action "takingphoto".

Figure 4: Some visualization results on Human3.6M dataset.

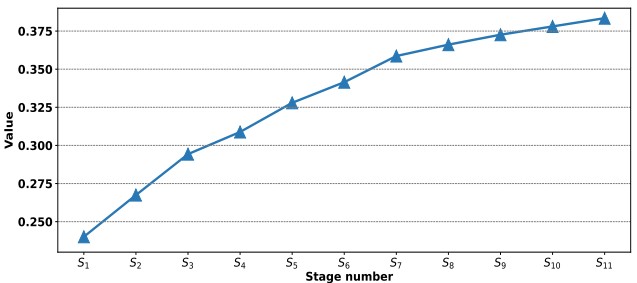

Figure 5: $\alpha$ at different stages.

the preservation of prior information, resulting in the forgetting of prior knowledge. Here, we utilize $\alpha$ to mitigate the loss of prior information, thereby improving the overall training of the model.

## 4.4 Ablation Studies

We conduct several ablation experiments to further verify the effectiveness of our proposed framework.

**Evaluation on the number of tasks.** As shown in Table 3, the model's performance improves as the number of tasks gets larger from 1 to 3, and it remains stable when the number of tasks becomes larger than 3.

Table 3: The average error of different numbers of tasks.

| number of tasks | 1 | 2 | 3 | 5 | 8 |
|---|---|---|---|---|---|
| avg error | 66.95 | 66.02 | 65.00 | 65.05 | 65.03 |

**Evaluation on different implementations.** As Figure 6 shows, we compare the results of four different implementations. PGBIG is our baseline model. "w/o $\alpha$" means that we only divide the training process into several stages to train each task without using the PCF. "HC" represents using a hand-crafted coefficient that changes its values similar to our PCF at each epoch. Specifically, in stage $S_1$, the value of $\alpha$ is set to 1. In stage $S_2$, $\alpha$ is initially set to 0.1 and increased by 0.05 at each epoch until reaching 0.5, where it remains constant. The same pattern applies to stage $S_3$. In each subfigure, we show the results of $Z_k$ on the validation set which is obtained at stage $S_k$.

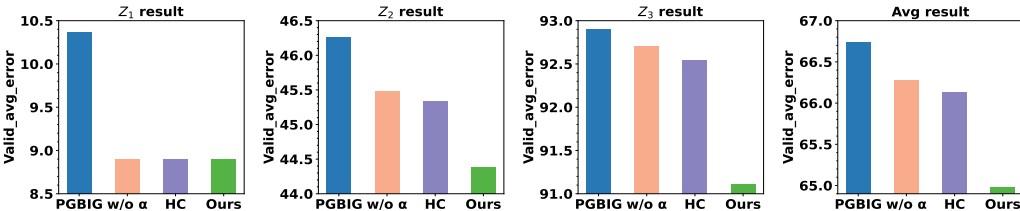

Figure 6: Comparison of different approaches. PGBIG is a baseline model that is trained without multi-stage. "w/o $\alpha$" represents training multi-stage process without PCF. "HC" means using a hand-designed coefficient. "Ours" is a multi-stage training process with PCF.

In stage $S_1$, where no prior information is available, the results of "w/o $\alpha$", "HC", and "Ours" are identical, but superior to the baseline. This validates the effectiveness of decomposing multiple-moment predictions, as it alleviates the constraint of long-term predictions on short-term predictions and enhances the model's ability to learn short-term predictions. In the following stages, the performance of "w/o $\alpha$" is worse than "Ours", which indicates that the prior knowledge exploited by our framework benefits the prediction model training. Moreover, as the training period progressed, the performance gap becomes larger. However, the method without $\alpha$ can still achieve better performance than "PGBIG", indicating that training model in a multi-stage manner can also exploit some useful prior information for prediction. We also note that the performance of "Ours" is much better than "HC", which conducts temporal continual learning with fixed and manually defined PCF. It demonstrates that joint training PCF and model parameters is beneficial.

**The forgetting of prior knowledge.** As shown in Table 4, introducing the prior compensation factor alleviates the performance degradation from stage $S_1$ to stage $S_3$ of task $Z_1$'s predictions. Specifically, without PCF, the prediction error of $Z_1$ increases by 0.83, whereas with PCF, it only increases by 0.27. This result suggests that PCF can effectively alleviate the forgetting issue. As a result, $Z_1$ can offer more comprehensive priors for $Z_2$ and $Z_3$ predictions, resulting in better prediction performance.

Table 4: The average error of different tasks at the end of each stage. $Z_i$ represents task $i$.

(a) Without PCF.

|       | $Z_1$ | $Z_2$ | $Z_3$ |
|-------|-------|-------|-------|
| $S_1$ | 9.03  | -     | -     |
| $S_2$ | 9.44  | 45.33 | -     |
| $S_3$ | 9.86  | 45.70 | 92.80 |

(b) Using PCF.

|       | $Z_1$ | $Z_2$ | $Z_3$ |
|-------|-------|-------|-------|
| $S_1$ | 9.03  | -     | -     |
| $S_2$ | 9.10  | 44.43 | -     |
| $S_3$ | 9.30  | 44.62 | 91.37 |

## 5   Conclusion

In this paper, we introduced the temporal continual learning framework for addressing the challenges in human motion prediction. Our framework addresses the constraint between long-term and short-term prediction, allowing for better utilization of prior knowledge from short-term prediction to enhance the performance of long-term prediction. Additionally, we introduced the prior compensation factor to mitigate the issue of forgetting information. Extensive experiments demonstrated our framework's effectiveness and flexibility.

**Limitation.** Our proposed training framework may slightly increase training time. However, the testing time remains unchanged compared to the backbone model.

**Broader Impact.** We believe our work has value for not only human motion prediction but also for more general prediction tasks and backbone models [41, 24, 16, 34]. This has benefits in various areas such as security monitoring, robotics, and autonomous driving.

## Acknowledgments and Disclosure of Funding

This work was supported partially by the NSFC (U21A20471, U22A2095, 62076260, 61772570), Guangdong Natural Science Funds Project (2020B1515120085, 2023B1515040025), Guangdong NSF for Distinguished Young Scholar (2022B1515020009), and Guangzhou Science and Technology Plan Project (202201011134).

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
