# Supplementary Material for Submission "Temporal Continual Learning with Prior Compensation for Human Motion Prediction"

**Jianwei Tang**
Sun Yat-sen University
tangjw7@mail2.sysu.edu.cn

**Jiangxin Sun**
Sun Yat-sen University
sunjx5@mail2.sysu.edu.cn

**Xiaotong Lin**
Sun Yat-sen University
linxt29@mail2.sysu.edu.cn

**Lifang Zhang**
Dongguan University of Technology
2017028@dgut.edu.cn

**Wei-Shi Zheng**
Sun Yat-sen University
wszheng@ieee.org

**Jian-Fang Hu**[*]
Sun Yat-sen University
hujf5@mail.sysu.edu.cn

## Abstract

In this supplementary document, we present additional details of the proof for the two lemmas discussed in our main submission. These details were excluded from the main submission due to space constraints. Additionally, we further include more experimental results of our approach on the Human3.6M dataset in this supplementary document.

In the main submission, we have presented a temporal continual learning (TCL) framework for human motion prediction, in which we need to optimize the following objective:

$$-\log P(Z_{1:k}; \boldsymbol{\theta}) = -\log P(Z_1; \boldsymbol{\theta}) - \sum_{i=2}^{k} \log(P(Z_i|\hat{Z}_{1:i-1}; \boldsymbol{\theta}) + \alpha_{Z_{1:i-1} \to Z_i}), \tag{1}$$

where $\alpha_{Z_{1:i-1} \to Z_i} \in [0, 1 - P(Z_i|\hat{Z}_{1:i-1}; \boldsymbol{\theta})]$ indicates the extent of forgotten knowledge when utilizing prior knowledge from tasks $Z_{1:i-1}$ to predict task $Z_i$. However, directly optimizing the objective is not easy. We have shown in the main submission that we can alternatively minimize its upper bound to obtain a reasonable solution for the TCL model, which can be formulated as:

$$
\begin{aligned}
UB = &-\log P(Z_1; \boldsymbol{\theta}) + \sum_{i=2}^{k} ((1 - \alpha_{Z_{1:i-1} \to Z_i})(-\log P(Z_i|\hat{Z}_{1:i-1}; \boldsymbol{\theta})) \\
&+ (1 - \alpha_{Z_{1:i-1} \to Z_i}) \log(1 - \alpha_{Z_{1:i-1} \to Z_i}) + \log(1 + \alpha_{Z_{1:i-1} \to Z_i})).
\end{aligned}
\tag{2}
$$

In the main submission, we have presented the following two Lemmas to show that the formulated upper bound is reasonable.

**Lemma 3.1.** For $0 \leq a \leq 1 - b$ and $0 < b \leq 1$, the inequality $-\log(a + b) \leq (1 - a)(-\log b) + (1 - a) \log(1 - a) + \log(1 + a)$ holds. The equality holds if and only if $a = 0$.

---

[*]Jian-Fang Hu is the corresponding author. Hu is also with Guangdong Province Key Laboratory of Information Security Technology, Guangzhou, China and Key Laboratory of Machine Intelligence and Advanced Computing, Ministry of Education, China.

37th Conference on Neural Information Processing Systems (NeurIPS 2023).

**Lemma 3.2.** The absolute difference between the target objective (Equation (1)) and upper bound (Equation (2)) is not larger than $\log(3/2) * (k - 1)$ when $P(Z_i | \hat{Z}_{1:i-1}; \boldsymbol{\theta}) \geq 1/2, i \in \{2, 3, \cdots, k\}$. This bound is achieved when $P(Z_k | \hat{Z}_{1:k-1}; \boldsymbol{\theta}) = 1/2$ and $\alpha_{Z_{1:i-1} \to Z_i} = 1/2, i \in \{2, 3, \cdots, k\}$.

In the following, we will provide detailed theoretical proof for the two lemmas.

## A  Theoretical Proof

### A.1  Proof of Lemma 3.1.

According to Jensen's inequality, for any real numbers $x_1$ and $x_2$ and non-negative weights $w_1$ and $w_2$ satisfying $w_1 + w_2 = 1$, if $H$ is a convex function, the following inequality holds:

$$H(w_1 x_1 + w_2 x_2) \leq w_1 H(x_1) + w_2 H(x_2). \tag{3}$$

The equality holds if and only if $x_1 = x_2$. Specifically, by setting $x_1 = x, x_2 = y, w_1 = 1/2$, $w_2 = 1/2$ ($x > 0$ and $y > 0$) and $H(u) = u \log(u)$, we can obtain a special case of Jensen's inequality:

$$\log \frac{x + y}{2} \leq \frac{x}{x + y} \log x + \frac{y}{x + y} \log y. \tag{4}$$

Then, considering $b \in (0, 1]$ and $a \in [0, 1 - b]$ and substituting $x = 1 - a$ and $y = a + b$ into Inequality (4), we can obtain the following inequality:

$$\log(a + b) \geq (1 - a - b) \log(a + b) + (1 + b) \log \frac{1 + b}{2} - (1 - a) \log(1 - a). \tag{5}$$

We can rewrite the right side of Inequality (5) as follows:

$$
\begin{aligned}
&(1 - a - b) \log(a + b) + (1 + b) \log \frac{1 + b}{2} - (1 - a) \log(1 - a) \\
&\geq (1 - a) \log b - b \log(a + b) + (1 + b) \log \frac{1 + b}{2} - (1 - a) \log(1 - a) \\
&\geq (1 - a) \log b - (1 - a) \log(1 - a) - \log(1 + a) + (1 + b) \log \frac{1 + b}{2} \\
&\geq (1 - a) \log b - (1 - a) \log(1 - a) - \log(1 + a) + \log \frac{1}{2}.
\end{aligned}
\tag{6}
$$

Here, we will show that even discarding the constant term $\log(1/2)$, the inequality still holds. Let's consider the following function:

$$G(a) = \log(a + b) - (1 - a) \log b + (1 - a) \log(1 - a) + \log(1 + a). \tag{7}$$

By defining $G_1(a) = \log(a + b) - (1 - a) \log b$ and $G_2(a) = (1 - a) \log(1 - a) + \log(1 + a)$, we can observe that $G(a) = G_1(a) + G_2(a)$.

Considering the function $G_1(a)$, we can calculate its derivative and second derivative as follows:

$$
\begin{aligned}
\dot{G}_1(a) &= \frac{1}{a + b} + \log b \\
\ddot{G}_1(a) &= -\frac{1}{(a + b)^2}.
\end{aligned}
\tag{8}
$$

We first consdier the case of $b \geq 1/e$. Since $\ddot{G}_1(a) < 0$ and $0 \leq a \leq 1 - b$, we can obtain that $\dot{G}_1(a) \geq \dot{G}_1(1 - b) = 1 + \log b \geq 0$. Therefore, $G_1(a) \geq G_1(0) = 0$.

For the case of $0 < b < 1/e$, we can easily obtain that $\dot{G}_1(1 - b) = 1 + \log b < 0$ and $\dot{G}_1(0) = 1/b + \log b > 0$. Hence, by considering the monotonicity of $\dot{G}_1(a)$, we can conclude that there exists an $a_0 \in (0, 1 - b)$ such that $\dot{G}_1(a) \geq 0$ for $a \in [0, a_0]$ and $\dot{G}_1(a) < 0$ for $a \in (a_0, 1 - b]$. As a result, we can get $G_1(a) \geq \min(G_1(0), G_1(1 - b))$. Since $G_1(0) = 0$ and $G_1(1 - b) = -b \log b \geq 0$, we can finally have $G_1(a) \geq 0$, the equality holds if only if $a = 0$.

Here, we show that $G_2(a) \geq 0$ holds. The first derivative $\dot{G}_2(a)$ and second derivative $\ddot{G}_2(a)$ are given by:

$$\dot{G}_2(a) = -\log(1-a) - 1 + \frac{1}{1+a}$$
$$\ddot{G}_2(a) = \frac{1}{1-a} - \frac{1}{(1+a)^2}. \tag{9}$$

Since $\ddot{G}_2(a) \geq 0$ for any $a \in [0, 1-b]$, we can get $\dot{G}_2(a) \geq \dot{G}_2(0) = 0$ and thus $G_2(a) \geq G_2(0) = 0$.

With $G_1(a) \geq 0$ and $G_2(a) \geq 0$, we can obtain $G(a) = G_1(a) + G_2(a) \geq 0$ and conclude that $-\log(a+b) \leq (1-a)(-\log b) + (1-a)\log(1-a) + \log(1+a)$. Moreover, we can observe that equality holds if and only if $a = 0$.

### A.2 Proof of Lemma 3.2.

For simplicity, let's denote $\alpha_i = \alpha_{Z_{1:i-1} \to Z_i}$ and $p_i = P(Z_i | \hat{Z}_{1:i-1}; \boldsymbol{\theta})$, where $p_i \in [1/2, 1], \alpha_i \in [0, 1-p_i], i \in \{2, 3, \cdots, k\}$. Then the difference between the upper bound (Equation (2)) and the target objective (Equation (1)) can be calculated as:

$$\Delta = UB - (-\log P(Z_{1:k}; \boldsymbol{\theta}))$$
$$= \sum_{i=2}^{k} \log(\alpha_i + p_i) + \sum_{i=2}^{k} ((1-\alpha_i)(-\log p_i) + (1-\alpha_i)\log(1-\alpha_i) + \log(1+\alpha_i)) \tag{10}$$
$$= \sum_{i=2}^{k} (\log(\alpha_i + p_i) - (1-\alpha_i)\log p_i + (1-\alpha_i)\log(1-\alpha_i) + \log(1+\alpha_i))).$$

Note that each term in the summation of Equation (10) has the same form, we denote it as $T(\alpha)$, where

$$T(\alpha) = \log(\alpha + p) - (1-\alpha)\log p + (1-\alpha)\log(1-\alpha) + \log(1+\alpha). \tag{11}$$

Then, we can turn to maximize $T(\alpha)$ in order to obtain the largest difference between the target objective and its upper bound.

The first derivative of $T(\alpha)$ can be calculated as:

$$\dot{T}(\alpha) = \frac{1}{\alpha + p} + \log p - \log(1-\alpha) - 1 + \frac{1}{1+\alpha}. \tag{12}$$

By defining $T_1(\alpha) = 1/(\alpha+p) + \log p$ and $T_2(\alpha) = -\log(1-\alpha) - 1 + 1/(1+\alpha)$, we can observe that $\dot{T}(\alpha) = T_1(\alpha) + T_2(\alpha)$. The first derivative of $T_2(\alpha)$ is:

$$\dot{T}_2(\alpha) = \frac{1}{1-\alpha} - \frac{1}{(1+\alpha)^2}. \tag{13}$$

Since $\dot{T}_2(\alpha) \geq 0$ for any $\alpha \in [0, 1-p]$, we get $T_2(\alpha) \geq T_2(0) = 0$. Furthermore, when $p \geq 1/2$, $T_1(\alpha) > 0$. Therefore, $\dot{T}(\alpha) = T_1(\alpha) + T_2(\alpha) > 0$, which means the maximum value of $T(\alpha)$ is:

$$T(1-p) = -p\log p + p\log p + (1-p) + \log(2-p)$$
$$= \log(2-p). \tag{14}$$

Since $p \geq 1/2$ and $\log(2-p)$ decreases with $p$, $T(1-p)$ will not larger than $\log(2 - 1/2) = \log(3/2)$. Particularly, when $p = 1/2$ and $\alpha = 1 - p = 1/2$, this largest bound will be achieved. Hence, we can conclude that $T(\alpha) \leq \log(3/2)$, and the equality holds when $p = \alpha = 1/2$.

Thus, as $\Delta = \sum_{i=2}^{k} T(\alpha_i)$ in Equation (10), we obtain $\Delta \leq \sum_{i=2}^{k} \log(3/2) = \log(3/2) * (k-1)$, which means the absolute difference between Equation (2) and Equation (1) will not be greater than $\log(3/2) * (k-1)$. The equality holds if and only if $p_i = \alpha_i = 1/2, i \in \{2, 3, \cdots, k\}$.

# B  More Experimental Results

## B.1  Detail experimental results

Table 1 displays the short-term prediction results for all action categories on the Human3.6M dataset. Table 2 shows the results for long-term prediction. Smaller values indicate better performance. It can be observed that our approach performs better than other competitors for the prediction of most of the actions. The improvement in short-term prediction demonstrates the effectiveness of the TCL framework in mitigating the constraint of long-term prediction on short-term prediction. The enhancement in long-term prediction also indicates that exploiting prior information depicted in the earlier stages with our proposed TCL framework is beneficial for long-term human motion prediction. We also note that our method achieves comparable results with PGBIG for the long-term prediction of a few action categories, such as "walking". This is because the action "walking" involves some repetitive motion patterns, which results in it being less important for mining prior motion information.

Table 1: Comparisons of short-term prediction on Human3.6M.

| action | walking | | | | eating | | | | smoking | | | | discussion | | | |
|---|---|---|---|---|---|---|---|---|---|---|---|---|---|---|---|---|
| ms | 80 | 160 | 320 | 400 | 80 | 160 | 320 | 400 | 80 | 160 | 320 | 400 | 80 | 160 | 320 | 400 |
| R.S. | 33.9 | 62.4 | 100.8 | 109.9 | 20.7 | 39.1 | 66.5 | 75.9 | 23.1 | 43.5 | 75.4 | 86.6 | 31.1 | 57.5 | 96.0 | 110.1 |
| LTD | 12.3 | 23.0 | 39.8 | 46.1 | 8.4 | 16.9 | 33.2 | 40.7 | 7.9 | 16.2 | 31.9 | 38.9 | 12.5 | 27.4 | 58.5 | 71.7 |
| PGBIG | 10.2 | 19.8 | 34.5 | 40.3 | 7.0 | 15.1 | 30.6 | 38.1 | 6.6 | 14.1 | 28.2 | 34.7 | 10.0 | 23.8 | 53.6 | 66.7 |
| PGBIG+Ours | **9.5** | **18.9** | **33.3** | **39.1** | **6.4** | **14.2** | **29.4** | **36.4** | **6.0** | **13.2** | **26.7** | **33.1** | **9.0** | **22.2** | **51.8** | **65.1** |
| action | directions | | | | greeting | | | | phoning | | | | posing | | | |
| ms | 80 | 160 | 320 | 400 | 80 | 160 | 320 | 400 | 80 | 160 | 320 | 400 | 80 | 160 | 320 | 400 |
| R.S. | 25.3 | 48.4 | 85.0 | 99.3 | 37.4 | 70.0 | 117.9 | 134.1 | 25.7 | 47.8 | 82.6 | 95.3 | 32.5 | 63.1 | 114.7 | 135.0 |
| LTD | 9.0 | 19.9 | 43.4 | 53.7 | 18.7 | 38.7 | 77.7 | 93.4 | 10.2 | 21.0 | 42.5 | 52.3 | 13.7 | 29.9 | 66.6 | 84.1 |
| PGBIG | 7.2 | 17.6 | 40.9 | 51.5 | 15.2 | 34.1 | 71.6 | 87.1 | 8.3 | 18.3 | 38.7 | 48.4 | 10.7 | 25.7 | 60.0 | 76.6 |
| PGBIG+Ours | **6.4** | **16.4** | **39.5** | **50.3** | **13.6** | **31.4** | **68.7** | **84.5** | **7.6** | **17.3** | **37.3** | **46.9** | **9.3** | **23.6** | **57.5** | **74.2** |
| action | purchases | | | | sitting | | | | sittingdown | | | | takingphoto | | | |
| ms | 80 | 160 | 320 | 400 | 80 | 160 | 320 | 400 | 80 | 160 | 320 | 400 | 80 | 160 | 320 | 400 |
| R.S. | 31.9 | 58.1 | 96.9 | 112.0 | 26.3 | 49.3 | 87.2 | 102.8 | 35.0 | 65.2 | 108.8 | 126.5 | 23.9 | 45.3 | 82.2 | 97.6 |
| LTD | 15.6 | 32.8 | 65.7 | 79.3 | 10.6 | 21.9 | 46.3 | 57.9 | 16.1 | 31.1 | 61.5 | 75.5 | 9.9 | 20.9 | 45.0 | 56.6 |
| PGBIG | 12.5 | 28.7 | 60.1 | 73.3 | 8.8 | 19.2 | 42.4 | 53.8 | 13.9 | 27.9 | 57.4 | 71.5 | 8.4 | 18.9 | 42.0 | 53.3 |
| PGBIG+Ours | **11.3** | **26.9** | **59.0** | **72.9** | **8.1** | **18.1** | **40.9** | **52.3** | **12.9** | **26.3** | **55.2** | **69.1** | **7.7** | **17.6** | **40.4** | **51.8** |
| action | waiting | | | | walkingdog | | | | walkingtogether | | | | average | | | |
| ms | 80 | 160 | 320 | 400 | 80 | 160 | 320 | 400 | 80 | 160 | 320 | 400 | 80 | 160 | 320 | 400 |
| R.S. | 29.4 | 55.1 | 96.7 | 112.6 | 43.9 | 78.1 | 121.0 | 135.9 | 27.9 | 52.1 | 86.4 | 96.4 | 29.9 | 55.7 | 94.5 | 108.7 |
| LTD | 11.4 | 24.0 | 50.1 | 61.5 | 23.4 | 46.2 | 83.5 | 96.0 | 10.5 | 21.0 | 38.5 | 45.2 | 12.7 | 26.1 | 52.3 | 63.5 |
| PGBIG | 8.9 | 20.1 | 43.6 | 54.3 | 18.8 | 39.9 | 73.7 | 86.4 | 8.7 | 18.6 | 34.4 | 41.0 | 10.3 | 22.7 | 47.4 | 58.5 |
| PGBIG+Ours | **7.9** | **18.4** | **41.0** | **51.6** | **17.3** | **37.1** | **72.2** | **85.5** | **8.2** | **17.7** | **33.0** | **39.2** | **9.4** | **21.3** | **45.7** | **56.8** |

Table 2: Comparisons of long-term prediction on Human3.6M.

| action | walking | | eating | | smoking | | discussion | | directions | | greeting | | phoning | | posing | |
|---|---|---|---|---|---|---|---|---|---|---|---|---|---|---|---|---|
| ms | 560 | 1000 | 560 | 1000 | 560 | 1000 | 560 | 1000 | 560 | 1000 | 560 | 1000 | 560 | 1000 | 560 | 1000 |
| R.S. | 117.5 | 133.3 | 90.7 | 123.3 | 102.3 | 133.9 | 134.2 | 168.4 | 118.2 | 150.5 | 158.4 | 197.0 | 114.4 | 152.1 | 168.8 | 229.0 |
| LTD | 54.1 | 59.8 | 53.4 | 77.8 | 50.7 | 72.6 | 91.6 | 121.5 | 71.0 | 101.8 | 115.4 | 148.8 | 69.2 | 103.1 | 114.5 | 173.0 |
| PGBIG | 48.1 | **56.4** | 51.1 | 76.0 | 46.5 | 69.5 | 87.1 | 118.2 | 69.3 | 100.4 | 110.2 | 143.5 | 65.9 | 102.7 | 106.1 | 164.8 |
| PGBIG+Ours | **47.1** | 56.9 | **48.8** | **74.1** | **44.4** | **66.7** | **86.2** | **117.2** | **68.4** | **99.6** | **108.8** | **142.7** | **64.0** | **100.0** | **103.9** | **163.9** |
| action | purchases | | sitting | | sitdown. | | takeph. | | waiting | | walkdog. | | walkto. | | average | |
| ms | 560 | 1000 | 560 | 1000 | 560 | 1000 | 560 | 1000 | 560 | 1000 | 560 | 1000 | 560 | 1000 | 560 | 1000 |
| R.S. | 136.2 | 172.9 | 129.0 | 176.9 | 159.8 | 209.5 | 124.4 | 175.7 | 134.9 | 173.0 | 159.3 | 198.8 | 107.0 | 126.5 | 130.1 | 168.0 |
| LTD | 102.0 | 143.5 | 78.3 | 119.7 | 100.0 | 150.2 | 77.4 | 119.8 | 79.4 | 108.1 | 111.9 | 148.9 | 55.0 | 65.6 | 81.6 | 114.3 |
| PGBIG | **95.3** | **133.3** | 74.4 | 116.1 | 96.7 | 147.8 | 74.3 | 118.6 | 72.2 | 103.4 | **104.7** | 139.8 | 51.9 | 64.3 | 76.9 | 110.3 |
| PGBIG+Ours | 95.5 | 134.9 | **72.6** | **113.8** | **94.3** | **146.1** | **72.5** | **114.6** | **70.0** | **102.7** | 104.8 | **139.0** | **49.0** | **59.8** | **75.4** | **108.8** |

## B.2   Zero-shot experiment

We also conduct a zero-shot experiment that trains the model on the Human3.6M dataset and tests on the re-aligned AMASS dataset. The results are tabulated in Table 3. As shown, our approach can still bring some benefits to the backbone model with a simple and direct implementation without a special model design, enabling the backbone model to adapt to the zero-shot learning setting.

Table 3: Results of the zero-shot experiment that trained on Human3.6M and tested on AMASS. A smaller value means a better result.

| ms | 80 | 160 | 320 | 400 | 560 | 1000 |
|---|---|---|---|---|---|---|
| PGBIG | **5.2** | **11.2** | 25.4 | 32.9 | 47.8 | 80.2 |
| PGBIG+Ours | **5.2** | **11.2** | **25.2** | **32.7** | **47.3** | **79.2** |