# OpenReview forum: "Temporal Continual Learning with Prior Compensation for Human Motion Prediction"
_NeurIPS.cc/2023/Conference — NeurIPS 2023 poster_

### Official Review · Reviewer_aXof · 2023-06-08

**Soundness:** 2 fair
**Presentation:** 2 fair
**Contribution:** 2 fair
**Rating:** 5
**Confidence:** 5

**Summary:**

The paper proposes a novel multi-stage training framework called Temporal Continual Learning (TCL) for Human Motion Prediction (HMP) to address the challenges of short-term and long-term predictions and the incorporation of prior information from past predictions into subsequent predictions. The Prior Compensation Factor (PCF) is introduced to compensate for the lost prior information, and an optimization objective is derived through theoretical derivation. The TCL framework can be easily integrated with different HMP backbone models and adapted to various datasets and applications. Extensive experiments on three HMP benchmark datasets demonstrate the effectiveness and flexibility of TCL.

**Strengths:**

+ Introducing the Temporal Continual Learning (TCL) framework, a multi-stage training framework that addresses the constraint between short-term and long-term prediction in Human Motion Prediction (HMP) and allows for better utilization of prior knowledge from short-term prediction to enhance the performance of long-term prediction.
+ Introducing the Prior Compensation Factor (PCF) to mitigate the issue of forgetting information during multi-stage training, and deriving a more reasonable optimization objective through theoretical derivation.
+ Exploring a new pipeline for the HMP task.

**Weaknesses:**

+ Writing:
  + Without periods in lines 113-117.
  + $P\left(Z_{k} \mid Z_{1} Z_{2} \cdots Z_{k-1} \theta\right)$ -> $P\left(Z_{k} \mid Z_{1} Z_{2} \cdots Z_{k-1}; \theta\right)$.
  + **The following concepts are confusing: short+long, short only, short then shorr long. The explanation in Figure 1 is not easy to understand. I suggest providing a figure to illustrate the difference between the three concepts.** It is essential to your motivation.
  + **Please provide more experimental settings about the toy example (Figure 1).**
  + Methods part should be carefully written. Otherwise, it is somewhat confusing.
+ It is not clear about the motivation for designing $\alpha$s. The authors name it Prior Compensation Factors in Section 3,2. Why can it compensate for forgotten knowledge when leveraging prior knowledge? It is a scalar and why can it reflect so much knowledge of complex motions?
+ For experiments:
  + Datasets: I suggest authors provide the results on AMASS datasets, which will make your work more solid.
  + Authors choose PGBIG  as the backbone. More choices and comparisons of backbones should be presented in the main result (not in ablation). This verifies the main contribution of the paper.
  + I did not find the codes and demo videos in the supplementary. (Not necessary. If provided, more convincing. Pretrained models with inference codes are acceptable.)


**Questions:**

+ It is clear that "our objective" is between the "actual objective" and "naive objective". In Figure 2, is "our objective" close to the "actual objective"? Why isn't it closer to the "naive objective"? Any intuition or proof?
+ Missing baselines or related work: [1, 2, 3, 4]. I would like to discuss with the authors about the choices of baselines.
+ Efficiency and multi-stage pipeline. Recent researches [5] suggest predicting motions in one stage, which is easier to train. Will the multi-stage training be harder to tune or train? Will it be more time-consuming? I would like to discuss with the authors about it.
+ It will be great if authors can discuss or provide zero-shot adaptation experiments on other datasets.



**I would like to discuss with the authors according to the authors' rebuttal. The experiments are not sufficient and the presentation of the paper is not good enough. However, the theoretical insights are interesting. Therefore, I provide a weak accept score here. I will adjust my score according to the authors' response. I will carefully check the authors' responses. I would like to discuss with the authors with details of the paper and improve the quality of the papers jointly.**



[1]: Zhong, Chongyang, et al. "Spatio-temporal gating-adjacency GCN for human motion prediction." Proceedings of the IEEE/CVF Conference on Computer Vision and Pattern Recognition. 2022.

[2]: Sofianos, Theodoros, et al. "Space-time-separable graph convolutional network for pose forecasting." Proceedings of the IEEE/CVF International Conference on Computer Vision. 2021.

[3]: Bouazizi, Arij, et al. "MotionMixer: MLP-based 3D Human Body Pose Forecasting."IJCAI 2022.

[4]: Guo, Wen, et al. "Back to mlp: A simple baseline for human motion prediction." Proceedings of the IEEE/CVF Winter Conference on Applications of Computer Vision. 2023.

[5]: Chen et al. "HumanMAC: Masked Motion Completion for Human Motion Prediction." arXiv preprint arXiv:2302.03665 (2023).

---


I revise my rating to borderline accept. See [detail](https://openreview.net/forum?id=v0GzRLvVp3&noteId=FWQ456OsEQ).

**Limitations:**

Authors discussed limitations.

---

> ### Author Rebuttal · Authors · 2023-08-10
>
> **Part 1 (Part 2 is in global rebuttal)**
>
> Thanks to the reviewer for the constructive comments. We have carefully addressed your concerns and provided detailed responses for each review.
>
> **Q1:Some issues with the wording.**
>
> Re: Thank you for pointing out them. We will correct them.
>
> **Q2: Some questions of fig.1.**
>
> Re: Thank you for your suggestion. "short+long" represents the training approach of the baseline model, where both short-term and long-term predictions are trained together. "short only" indicates that only short-term predictions are trained on the baseline model without considering long-term predictions. "short then short+long" means that after training the model using the "short only" approach, the model is further trained by combining both short-term and long-term predictions together. We will provide a figure to illustrate the difference between different concepts and express it in a clearer manner. As for the experimental setting of Fig. 1, we followed the implementation details of the backbone PGBIG.
>
> **Q3: It is not clear about the motivation for designing αs. The authors name it Prior Compensation Factors in Section 3,2. Why can it compensate for forgotten knowledge when leveraging prior knowledge? It is a scalar and why can it reflect so much knowledge of complex motions?**
>
> Re: As we conduct a preliminary experiment and find that knowledge learned in short-term prediction can serve as a prior to facilitate the learning of far-future, this motivates us to formally model the motion prediction and propose a multi-stage training strategy to better exploit this prior knowledge. However, since we discover that the prior knowledge acquired from previous tasks diminishes when switching stages, our motivation for designing αs is to address this issue and estimate this lost prior knowledge to assist in predicting subsequent task. By applying a series of derivations, we obtain the final objective function as Eq. 5, where αs are formally presented as controlling the weighted combination of losses at different stages. Since the fitting of long-term prediction dominates the learning process, the prediction model struggles to preserve the prior knowledge learnt from previous short-term predictions. Consequently, the loss function assigns lower weights to current task to mitigate the loss of forgotten information. Thus, αs which is a vector does play a compensatory role for the forgotten knowledge and demonstrates effectiveness.
>
> **Q4: Datasets: I suggest authors provide the results on AMASS datasets, which will make your work more solid.**
>
> Re: We follow your suggestion and conduct experiment on the AMASS dataset. However, due to the unavailability of some sub-dataset ("Eyes_Japan_Dataset" and "BioMotionLab_NTroje") in AMASS, we could only perform experiments on its subset. We use the STSGCN [2] model you mentioned as backbone. Detailed results can be found in Q8. The experimental results indicate that our strategy effectively enhances model training on this dataset.
>
> **Q5: Authors choose PGBIG as the backbone. More choices and comparisons of backbones should be presented in the main result (not in ablation). This verifies the main contribution of the paper.**
>
> Re: We select the PGBIG as the backbone of our main experiments as it is the state-of-the-art architectures for human motion prediction under the common setting. However, we have also followed your suggestion to test the results of our approach with other backbones such as STSGCN [2] (GCN-based), MotionMixer [3] (MLP-based), siMLPe [4] (MLP-based) and POTR [6] (Transformer-based). The results can be found in Q8. As can be seen, our training method still improves the performance of the original prediction backbone model.
>
> **Q6: I did not find the codes and demo videos in the supplementary. (Not necessary. If provided, more convincing. Pretrained models with inference codes are acceptable.)**
>
> Re: We have provided a link of demo videoto AC. Due to NIPS' official policy that prohibits providing external links in the rebuttal, we would release our codes and pre-trained models after this work is accepted for publication.
>
> **Q7: It is clear that "our objective" is between the "actual objective" and "naive objective". In Figure 2, is "our objective" close to the "actual objective"? Why isn't it closer to the "naive objective"? Any intuition or proof?**
>
> Re: Fig. 2 is a illustrative diagram illustrating the relative positioning among "naive objective", "our objective" and "actual objective". This serves to demonstrate that our objective is a superior upper bound compared to the naive approach.
>
> **Q8: Missing baselines or related work: [1, 2, 3, 4]. I would like to discuss with the authors about the choices of baselines.**
>
> Re: Our approach can be flexibly applied to various backbones, such as [1, 2, 3, 4]. [1] enhances the generalization ability of GCN using a gating network. [2] utilizes space-time separable GCN to extract features from different dimensions. [3] presents an MLP-based architecture that effectively leverages spatiotemporal aggregated features. [4] is a recent MLP-based model. Due to the lack of open-source code for [1], we can only report experimental results for [2], [3] and [4]. The experiment of baseline [2] was conducted on the AMASS dataset with mean per joint position error as the evaluation metric. The experiments of baselines [3, 4] were conducted on the Human3.6M dataset, and the evaluation metric used was mean per joint position error. We also conducted experiments on POTR [6] (a Transformer-based model). And the results for POTR are shown using euler angle error as the evaluation metric. From the experimental results in the Table 6, 7, 9 and 10, it can be observed that our strategy consistently improves the performance of these backbones, validating the effectiveness of our proposed strategy. We will provide discussions on [1, 2, 3, 4] in subsequent versions of the manuscript.

---

> > ### Comment · Reviewer_aXof · 2023-08-10
> > **Response#1 to authors (after rebuttal)**
> >
> > Thanks for your efforts. My concerns still exist.
> >
> > I reply to the author's rebuttal ASAP to allow more time for the author's feedback.
> >
> > + For Re1, please show how you will revise the method part. How can I be convinced?
> >
> > + Authors did not provide a figure in the rebuttal pdf. It is not convincing.
> >
> > + I still do not know what $\alpha$s mean. "Thus, $\alpha$s which is a vector does play a compensatory role for the forgotten knowledge and demonstrates effectiveness." Are $\alpha$s vectors or scalars?
> >
> > + Note that "Eyes_Japan_Dataset" and "BioMotionLab_NTroje" are used in the HumanML3D dataset. Besides, [2] provides experiments on AMASS (author acknowledged in rebuttal). I am curious about why they cannot be used.
> >
> > + For Re5, the author did not get my idea. My comment is that the ablation should be your main table result to verify your claim.
> >
> > + For Re6, I will discuss with AC in the following process.
> >
> > + **Re7 is dodging my question.** In Figure 2, is "our objective" close to the "actual objective"? Why isn't it closer to the "naive objective"? Any intuition or proof? Can you provide any answer?
> >
> > + For Re8, will you release the code about mentioned experiments if accepted? Note that both reviews and responses will be open if accepted.
> >
> > + The zero-shot experiment is not convinced enough. The train and zero-shot experiments are both on the Human3.6M dataset. It seems no challenges for the method. A better choice is training on H3.6M and test in AMASS. For code and the setting, please refer to `https://github.com/LinghaoChan/HumanMAC#zero-shot-prediction-on-amass`. Therefore, I will not improve my score according to current experiments.
> >
> >
> >
> > ### I read other reviewers' concern during the rebuttal and provide following concerns.
> > + I found reviewer skai show the same concern on $\alpha$s. And author did not provide the figure suggested by reviewer skai. Waht is the reason.
> >
> > + "Q6 by GqjD: In Eq. 5 the weights for each term is changed from \alpha to 1-\alpha, is this really valid?" I have the same concern with reviewer GqjD. Author did not provide any evidence to verify it.

---

> > > ### Author Response · Authors · 2023-08-14
> > > **Replying to the reviewer's comments (Part 1)**
> > >
> > > Thanks to the reviewer for the constructive comments.
> > >
> > > **Q1: For Re1, please show how you will revise the method part. How can I be convinced?**
> > >
> > > Re: We will revise the method part mainly in the following aspects. For the expressions in the training periods, we will explicitly define the human motion prediction problem, which aims to predict the future sequence with a period  $T_h+1:T_{Z_K}$ conditioned on the observed history $T_1:T_h$. To achieve this, we first divide the future sequences into K segments, with each segment denoted as k ranging from $T_{Z_{k-1}}+1$ to $T_{Z_k}$. And the task $Z_k$ is defined to predict the motions of the k-th segment. We will include this in the revision. As for the ';' before $\theta$, we will follow your suggestion and rewrite it. Also, we accidentally omit an explanation in Eq. (3) where α is a random variable. With this explanation, it would be clearer about the form of α in the loss function. We hope the revision is now acceptable.
> > >
> > > **Q2: Authors did not provide a figure in the rebuttal pdf. It is not convincing.**
> > >
> > > Re: Actually, the pdf is already filled with experiment result tables as required by the reviewers which occupy an entire page. Thus limited by the space constraints (1 page), we are unable to include a figure in the rebuttal pdf. Note that we have already explained Fig. 1 in our initial rebuttal response with three statements ("short only" indicates that only short-term predictions are trained on the baseline model without considering long-term predictions. "short then short+long" means that after training the model using the "short only" approach, the model is further trained for both short-term and long-term predictions jointly), which is considered to be a clear demonstration for the training strategy. In our revision, we will involve these in the figure and explicitly illustrate the sequences and strategy involved in the toy experiments.
> > >
> > > **Q3: I still do not know what αs mean. "Thus, αs which is a vector does play a compensatory role for the forgotten knowledge and demonstrates effectiveness." Are αs vectors or scalars?**
> > >
> > > Re: α is a scalar. Here, αs refer to a vector containing multiple elements and the i-th element is a scalar for the i-th training sample trained in the certain stage.
> > >
> > > **Q4: Note that "Eyes_Japan_Dataset" and "BioMotionLab_NTroje" are used in the HumanML3D dataset. Besides, [2] provides experiments on AMASS (author acknowledged in rebuttal). I am curious about why they cannot be used.**
> > >
> > > Re: We have double checked and confirm that the link to the AMASS dataset provided in [2] does not include the BioMotionLab_NTroje and Eyes_Japan_Dataset subset. We suspect that the subset is not available due to privacy or copyright concerns, since [2] is an early work published before 2021. Indeed, this is not a special case in the community. For example, raw videos in Human3.6M are no longer available, even though they could be downloaded before 2022.
> > >
> > > **Q5: For Re5, the author did not get my idea. My comment is that the ablation should be your main table result to verify your claim.**
> > >
> > > Re: Thank you for your suggestion. In the ablation study, we have provided the results on all of the datasets involved in the main experiment section. We will move the results to the main table in the final version.
> > >
> > > **Q6: Re7 is dodging my question. In Figure 2, is "our objective" close to the "actual objective"? Why isn't it closer to the "naive objective"? Any intuition or proof? Can you provide any answer?**
> > >
> > > Re: Indeed, Figure 2 is provided as a simple graphic illustration showing that our objective is closer to "actual objective" than "naive objective". With respect to your concern that "our objective" is closer to "actual objective" or "naive objective", we cannot provide a definite conclusion about which is closer, as the distances depend on how well the model learned. We would like to further clarify that this is not our main concern and it does not affect our conclusion derived in this work.
> > >
> > > **Q7: For Re8, will you release the code about mentioned experiments if accepted? Note that both reviews and responses will be open if accepted.**
> > >
> > > Re: Yes. We will release code of all the experiments if accepted.

---

> > > > ### Author Response · Authors · 2023-08-14
> > > > **Replying to the reviewer's comments (Part 2)**
> > > >
> > > >
> > > >
> > > > **Q8: The zero-shot experiment is not convinced enough. The train and zero-shot experiments are both on the Human3.6M dataset. It seems no challenges for the method. A better choice is training on H3.6M and test in AMASS. For code and the setting, please refer to HumanMAC.**
> > > >
> > > > Re: Thank you for the suggestion. We notice that HumanMAC employs a realignment tool to re-align the joint annotations of the AMASS with the Human3.6M. We have followed the implementations in HumanMAC and utilized the re-aligned AMASS dataset to conduct the zero-shot experiment. The results are tabulated in the following. As shown, our approach can still bring some benefits to the PGBIG model with a simple and direct implementation without a special model design, enabling our framework to adapt to the zero-shot learning setting.
> > > >
> > > > |            | 80ms | 160ms | 320ms | 400ms | 560ms | 1000ms |
> > > > | ---------- | ---- | ----- | ----- | ----- | ----- | ------ |
> > > > | PGBIG      | 5.2  | 11.2  | 25.4  | 32.9  | 47.8  | 80.2   |
> > > > | PGBIG+Ours | 5.2  | 11.2  | 25.2  | 32.7  | 47.3  | 79.2   |
> > > >
> > > > **Q9: I found reviewer skai show the same concern on αs. And author did not provide the figure suggested by reviewer skai. What is the reason.**
> > > >
> > > > Re: Due to space limitation and the rebuttal pdf being filled with experiment result tables as requested by the reviewers, we are not able to provide a figure in the rebuttal pdf. We would like to point out that reviewer skai suggests a different way to address the loss function/model derived in our manuscript, in which a closed-form α solution is obtained in the optimization of each training stage for each individual sample. While in our implementation, we explicitly embed the α learning into our framework and model it as a network output, calculating it in the data-driven manner. Visually, the α obtained with the code provided by reviewer skai has a similar trend to that obtained by our approach. However, our approaches performs better than the suggested closed-form α solution (65.0 vs. 65.5 in the term of mean MPJPE). Please refer to Table 1 in the rebuttal pdf for more detailed results. We will include the discussions in the supplementary materials.
> > > >
> > > > **Q10: "Q6 by GqjD: In Eq. 5 the weights for each term is changed from \alpha to 1-\alpha, is this really valid?" I have the same concern with reviewer GqjD. Author did not provide any evidence to verify it.**
> > > >
> > > > Re: Yes, it is valid. We would like to clarify that the α in Eq. (4) is not a weight, it is a variable in the logarithm operation. By using Lemma 3.1 (Lines 155-156 in the main manuscript), we can obtain Eq. (5) in which the 1-α acts as a combination weight to aggregate the losses achieved for different tasks. The detailed theoretical derivation can be seen in Lines 21-47 in our supplemental material.

---

> > > > ### Comment · Reviewer_aXof · 2023-08-14
> > > > **Re: Replying to the reviewer's comments**
> > > >
> > > > I acknowledge the authors' efforts (Re1, Re5, Re7, Re8). I will list the remaining questions.
> > > >
> > > > For Re2 and Re9, the rebuttal is lack of caption. It is really hard to find the corresponding result. For Re2, please add figure to explain it. Otherwise, it will be really confusing.
> > > >
> > > > For Re3, I am still confusing why a scalar can represent a motion. Authors can provide some intuitive explanations and I will also discuss it with other reviewers.
> > > >
> > > > For Re4, I am still confused. HumanML3D contains the subset. I am using it now. Please check the HumanML3D's code: `https://github.com/EricGuo5513/HumanML3D/blob/main/raw_pose_processing.ipynb`. The codes include the processing of `Eyes_Japan_Dataset `.
> > > >
> > > > For Re6, it will not affect the result. However, it might mislead readers. My suggestion is to revise the figure. Otherwise, this may contain hints of over-claim.
> > > >
> > > > Are there any experiments to support Re10?

---

> > > > > ### Author Response · Authors · 2023-08-16
> > > > > **Replying to the reviewer's comments**
> > > > >
> > > > > Thank you for your constructive comments.
> > > > >
> > > > > **Q1: For Re2 and Re9, the rebuttal is lack of caption. It is really hard to find the corresponding result. For Re2, please add figure to explain it. Otherwise, it will be really confusing.**
> > > > >
> > > > > Re: Sorry for the lack of captions in the rebuttal pdf. Since the pdf is not editable now, we have to provide the captions in the following.
> > > > >
> > > > > Table 1. Comparison of HC2 and ours.
> > > > >
> > > > > Table 2. Results of different number of stages.
> > > > >
> > > > > Table 3. The average error of different tasks at different stages.
> > > > >
> > > > > Table 4. Comparison of stage-wise $\alpha$ and ours.
> > > > >
> > > > > Table 5. Comparison of the cost of training time.
> > > > >
> > > > > Table 6. Comparison using siMLPe as backbone.
> > > > >
> > > > > Table 7. Comparison using POTR as backbone.
> > > > >
> > > > > Table 8. Comparison of simple dynamic loss and ours.
> > > > >
> > > > > Table 9. Comparison using STSGCN as backbone.
> > > > >
> > > > > Table 10. Comparison using MotionMixer as backbone.
> > > > >
> > > > > Regarding the figure for Re2, we have followed your suggestion and provided an anonymous link to the AC, which includes a figure to indicate the differences among "short only", "short+long" and "short then short+long". Hope it is acceptable now.
> > > > >
> > > > > **Q2: For Re3, I am still confusing why a scalar can represent a motion. Authors can provide some intuitive explanations and I will also discuss it with other reviewers.**
> > > > >
> > > > > Re: We would like to clarify that $\alpha$ is not a representation of motion. It is defined as the information gap between $P(Z_{k}|Z_{1}Z_{2}\cdots Z_{k-1};\theta)$ and $P(Z_{k}|\hat{Z}_{1:k-1};\theta)$, which has different values for different samples at varied training tasks and is learned by the model adaptively. We have theoretically shown in Eq. (5) that the learned $\alpha$s can finally act as weights (scalars) to balance the training losses achieved for different training samples and tasks. These $\alpha$s work together to prevent the model from forgetting the prior knowledge, resulting in better performances in all of our experiments. Hope it is clear now to show how $\alpha$ works in the proposed temporal continual learning framework.
> > > > >
> > > > > **Q3: For Re4, I am still confused. HumanML3D contains the subset. I am using it now. Please check the HumanML3D's code: `https://github.com/EricGuo5513/HumanML3D/blob/main/raw_pose_processing.ipynb`. The codes include the processing of `Eyes_Japan_Dataset `.**
> > > > >
> > > > > Re: Thank you for your helpful suggestion. We have followed the instruction of HumanML3D to obtain the subset and conducted experiments using STSGCN as our backbone. The results are presented in the table below. It can be observed that our training strategy can consistently improve the performance again on this set. We could include the results in our final version if required.
> > > > >
> > > > > |                 | **80ms⬇** | **160ms⬇** | **320ms⬇** | **400ms⬇** | **560ms⬇** | **1000ms⬇** |
> > > > > | :-------------: | :-------: | :--------: | :--------: | :--------: | :--------: | :---------: |
> > > > > |   **STSGCN**    |   12.53   |   26.33    |   53.32    |   63.99    |   80.27    |   104.21    |
> > > > > | **STSGCN+Ours** | **10.98** | **24.38**  | **51.33**  | **61.79**  | **78.13**  | **102.86**  |
> > > > >
> > > > > **Q4: For Re6, it will not affect the result. However, it might mislead readers. My suggestion is to revise the figure. Otherwise, this may contain hints of over-claim.**
> > > > >
> > > > > Re: Sorry for the confusion caused by the figure. We would provide some clarification about it in the figure caption in the final version.
> > > > >
> > > > > **Q5: Are there any experiments to support Re10?**
> > > > >
> > > > > Re: Unfortunately, we are unable to report experimental results about Eq. (4) as it cannot be directly optimized. However, we have theoretically shown that with Lemma 3.1, Eq. (4) can be relaxed into Eq. (5), which can be optimized directly. Please let us know if we have misunderstood your concern.

---

> > > > > > ### Comment · Reviewer_aXof · 2023-08-17
> > > > > > **Re: Replying to the reviewer's comments**
> > > > > >
> > > > > > When you used the HumanML3D guidance, did you use all AMASS datasets? Because HumanML3D is based on AMASS. Do you use the 263-dim (per frame) motion data?
> > > > > >
> > > > > > For the previous discussion, are problems on "Eyes_Japan_Dataset" and "BioMotionLab_NTroje" resolved?

---

> > > > > > > ### Author Response · Authors · 2023-08-17
> > > > > > > **Replying to the reviewer's comments**
> > > > > > >
> > > > > > >
> > > > > > >
> > > > > > > **Q1: When you used the HumanML3D guidance, did you use all AMASS datasets? Because HumanML3D is based on AMASS.**
> > > > > > >
> > > > > > > Re: In this experiment, we exactly followed implementations in STSGCN [1], where only 13 subsets from AMASS were used (8 for training, 4 for validation and 1 for testing). The subsets 'BMLmovi', 'BMLhandball', 'Transitions', 'SSM' and 'DFaust' are not used in the experiment. We have double checked that the approaches in [2,3] have also followed this training-validation-testing setting. The detailed dataset split for training/validation/testing is listed as follows:
> > > > > > >
> > > > > > > ```
> > > > > > > training set:['CMU', 'MPI_Limits', 'TotalCapture', 'Eyes_Japan_Dataset', 'KIT', 'EKUT', 'TCD_handMocap', 'ACCAD']
> > > > > > > validation set:['HumanEva', 'MPI_HDM05', 'SFU', 'MPI_mosh']
> > > > > > > testing set:['BioMotionLab_NTroje']
> > > > > > > ```
> > > > > > >
> > > > > > > More detailed validation and testing results are tabulated in the following. As can be seen, our training strategy can consistently improve the performance again on this set.
> > > > > > >
> > > > > > > ​									**Table 1. Results on validation and testing sets of AMASS dataset**
> > > > > > >
> > > > > > > |                              | **80ms⬇** | **160ms⬇** | **320ms⬇** | **400ms⬇** | **560ms⬇** | **1000ms⬇** |
> > > > > > > | ---------------------------- | --------- | ---------- | ---------- | ---------- | ---------- | ----------- |
> > > > > > > | **STSGCN (validation)**      | 12.53     | 26.33      | 53.32      | 63.99      | 80.27      | 104.21      |
> > > > > > > | **STSGCN+Ours (validation)** | 10.98     | 24.38      | 51.33      | 61.79      | 78.13      | 102.86      |
> > > > > > > | **STSGCN (testing)**         | 11.39     | 20.78      | 37.16      | 43.78      | 53.76      | 69.66       |
> > > > > > > | **STSGCN+Ours (testing)**    | **10.88** | **20.23**  | **36.65**  | **42.83**  | **52.35**  | **68.11**   |
> > > > > > >
> > > > > > > **Q2: Do you use the 263-dim (per frame) motion data?**
> > > > > > >
> > > > > > > Re: No, we have exactly followed the implementations provided by STSGCN [1] and employed the 54-dim (per frame) motion data, where each human pose is represented by 18 body joints, and each joint is indicated by the 3-dim x-y-z coordinates. We simply feed the 54-dim data into the STSGCN and STSGCN+ours models, as done in [1,2,3].
> > > > > > >
> > > > > > > **Q3: For the previous discussion, are problems on "Eyes_Japan_Dataset" and "BioMotionLab_NTroje" resolved?**
> > > > > > >
> > > > > > > Re: Yes. At first, we didn't find the subsets because their names had been changed in the AMASS official website (e.g., 'BioMotionLab_NTroje' is renamed as 'BMLrub'). But thanks to the link you provided previously, we have successfully downloaded all the subsets.
> > > > > > >
> > > > > > > **Reference:**
> > > > > > >
> > > > > > > [1] Sofianos, T., Sampieri, A., Franco, L., & Galasso, F. (2021). Space-time-separable graph convolutional network for pose forecasting. In Proceedings of the IEEE/CVF International Conference on Computer Vision (pp. 11209-11218).
> > > > > > >
> > > > > > > [2] Bouazizi, A., Holzbock, A., Kressel, U., Dietmayer, K., & Belagiannis, V. (2022). Motionmixer: Mlp-based 3d human body pose forecasting. arXiv preprint arXiv:2207.00499.
> > > > > > >
> > > > > > > [3] Guo, W., Du, Y., Shen, X., Lepetit, V., Alameda-Pineda, X., & Moreno-Noguer, F. (2023). Back to mlp: A simple baseline for human motion prediction. In Proceedings of the IEEE/CVF Winter Conference on Applications of Computer Vision (pp. 4809-4819).

---

> > > > > > > > ### Comment · Reviewer_aXof · 2023-08-18
> > > > > > > >
> > > > > > > > For previous refinements and all new experiments, I suggest authors revise the paper accordingly if accepted.
> > > > > > > >
> > > > > > > > For Re3, I can find `EyesJapanDataset` on the AMASS web page. What causes it?

---

> > > > > > > > > ### Author Response · Authors · 2023-08-18
> > > > > > > > > **Replying to the reviewer's comments**
> > > > > > > > >
> > > > > > > > > **Q1: For previous refinements and all new experiments, I suggest authors revise the paper accordingly if accepted.**
> > > > > > > > >
> > > > > > > > > Re: Thank you for your helpful suggestions to improve our manuscript. We will include all the refinements and new experiments in the final version.
> > > > > > > > >
> > > > > > > > > **Q2: For Re3, I can find `EyesJapanDataset` on the AMASS web page. What causes it?**
> > > > > > > > >
> > > > > > > > > Re: We previously failed to find the 'EyesJapanDataset' by searching the texts defined in the code provided by STSGCN, in which it is provided in the form of 'Eyes_Japan_Dataset' as shown in the following code. We apologize for missing the set due to our carelessness in our initial response.
> > > > > > > > >
> > > > > > > > > ```python
> > > > > > > > > class Datasets(Dataset):
> > > > > > > > >     def __init__(...):
> > > > > > > > >         ...
> > > > > > > > > 		amass_splits = [
> > > > > > > > >             ['CMU', 'MPI_Limits', 'TotalCapture', 'Eyes_Japan_Dataset', 'KIT', 'EKUT', 'TCD_handMocap', 'ACCAD'],
> > > > > > > > >             ['HumanEva', 'MPI_HDM05', 'SFU', 'MPI_mosh'],
> > > > > > > > >             ['BioMotionLab_NTroje'],
> > > > > > > > >         ]
> > > > > > > > > ```

---

> > > > > > ### Comment · Area_Chair_ewXw · 2023-08-17
> > > > > >
> > > > > > Regarding: "The following concepts are confusing: short+long, short only, short then short long. The explanation in Figure 1 is not easy to understand. I suggest providing a figure to illustrate the difference between the three concepts."
> > > > > >
> > > > > > The authors provided a figure in form of a video:
> > > > > >
> > > > > > https://youtube.com/shorts/tMqao8uNwP0

---

> > > > > > > ### Comment · Reviewer_aXof · 2023-08-17
> > > > > > > **About Fig. 1**
> > > > > > >
> > > > > > > Thanks to AC and the authors! It resolved my concern. I suggest authors revise the paper accordingly if accepted.

---

> > ### Comment · Reviewer_aXof · 2023-08-14
> >
> > If authors finished resolving some sub-questions, you can provide sub-responses first.

---

> ### Comment · Reviewer_aXof · 2023-08-18
> **Revise my rating**
>
> Dear AC, reviewers, and authors,
>
> I provide my latest rating here. I have carefully read the manuscript, supplementary, authors’ rebuttal, and other reviews. Therefore, I provide my comment for rebuttal and other reviews [here](https://openreview.net/forum?id=v0GzRLvVp3&noteId=7H0dti0atS). After many rounds of discussion, I have a better understanding of this manuscript.
>
> During rebuttal and discussion, the authors try to resolve my concerns. **Following my code-level suggestions, the authors provide zero-shot experiments on AMASS, training, and prediction results on AMASS.** I appreciate the authors' efforts. Besides, the authors took my advice and detailed how to revise the manuscript if accepted. This is serious and convincing. And experiments of this work are more solid than the first version. However, some of my concerns still exist.
>
> **This manuscript exist some significant problems on writing in the submission.** The caption in Fig. 1 is really hard to follow in the first version. $\alpha$s is confusing for readers (reviewer skai presents this concern at first as well). In the rebuttal pdf, there is no caption for tables. It seems like a not well-prepared submission.
>
> When performing experiments on AMASS, authors claimed multiple times ([first time in Re4](https://openreview.net/forum?id=v0GzRLvVp3&noteId=JXNkeD6aGJ), [second time in Re4](https://openreview.net/forum?id=v0GzRLvVp3&noteId=aMZNvA21Lp)) that the data could not be found. After I provide the authors a [guidance](https://openreview.net/forum?id=v0GzRLvVp3&noteId=gktdoOx4BG) for downloading the data, the author finally found the data ([See Re3](https://openreview.net/forum?id=v0GzRLvVp3&noteId=4grqIk0xER)). **The authors also acknowledged the carelessness during the [discussion](https://openreview.net/forum?id=v0GzRLvVp3&noteId=GCaELD7JRr).** Since AMASS (cited by 650+) is a dataset commonly used by peers and community who study human motions, **I have concerns about the professionalism of the author**.
>
> Overall, I think the quality of this work is at **the bottom ~10% of accepted NeurIPS papers**. That is to say, it is **marginally $\underline{\text{above}}$ the borderline**. Therefore, I will revise my rating **from WEAK ACCEPT to BORDERLINE ACCEPT confidently**.
>
> Best,
>
> Reviewer aXof

---

> > ### Author Response · Authors · 2023-08-19
> > **Replying to the reviewer's comments**
> >
> > Thanks for appreciating our efforts at this work. **Indeed, the contribution of this work is significant and its effectiveness has been demonstrated by extensive experiments, including the experiments on three datasets (Human3.6M, CMU-MoCap and 3DPW) reported in the manuscript and the new experiment on AMASS dataset required during rebuttal.** We would follow your suggestion to improve our writing in the revision, including the statement of $\alpha$s and the caption in Figure 1. We would appreciate it if you could consider how our work inspires and helps the future research in the community.

---

> > > ### Comment · Reviewer_aXof · 2023-08-19
> > >
> > > My recommendation for accepting this paper is not as strong as before (pre-rebuttal). After the discussion, I still think this work is above the borderline (>=5). Therefore, I changed the rating. I hope the author can understand my decision.
> > >
> > > I appreciate the authors' discussions on how this work inspires and helps future research in the community.  My suggestions:
> > >
> > > + Release more experiments on Human3.6M, CMU-MoCap, AMASS, and 3DPW when releasing the codes.
> > >
> > > + Present the zero-shot experiments when releasing the codes.
> > >
> > > + Authors can provide a demo video to present the strength of this work.
> > >
> > > + Provide detailed guidance for peers. This will improve your reputation and attract more follow-up research.
> > >
> > > Am I misunderstanding something?

---

### Official Review · Reviewer_vWPd · 2023-06-23

**Soundness:** 2 fair
**Presentation:** 2 fair
**Contribution:** 2 fair
**Rating:** 5
**Confidence:** 3

**Summary:**

This paper introduces the continual learning insight into human motion prediction. By analysis of the performance relationship between the short and long-term prediction, a compensatory method is proposed in a multi-stage learning setting. On several widely-used benchmarks, the proposed method is cooperated with different backbones and methods and compared with previous methods. Some decent progress is shown according to the analyses.

**Strengths:**

+ Splitting the long-term prediction into multi-stage and using the continuous learning insight is interesting and non-trivial.

+ The method proposal looks sound and designed well.

+ A good comparison and discussion are given to support the proposed method.

**Weaknesses:**

- What is the additional cost of using the proposed method? Please discuss the efficiency and the other possible cost.

- Lacking a vivid figure to illustrate the whole method pipeline before the detailed method introduction. Besides, the introduction part, especially the method description can be organized better and more logically.

- L128-130: Though embedding the “knowledge” into the trained parameters seems very reasonable, this is just an intuitive discussion. A more detailed and clear explanation is essential to illustrate what is knowledge really, or what sign we can get to observe the knowledge utilized or not, to avoid a metaphysical or empirical discussion only.

- Fig. 3: hard to read and discover the difference between methods.

- Lacking direct and clear experiments to show the effectiveness of the continual learning design, e.g., better avoidance of forgetting, better balance of the shot and long predictions, etc.

- typo: Fig. 1: lacking space before the (

**Questions:**

1. Though there are some "traditional" settings of the length definition of the long and short terms. I wonder if we change the 5 and 15 frames setting, is the analysis of Fig. 1 kept or changed?

2. Learning rate in Eq. 9: choice discussion.

**Limitations:**

Please add a discussion.

---

> ### Author Rebuttal · Authors · 2023-08-10
>
> Thanks to the reviewer for the constructive comments. We have carefully addressed your concerns and provided detailed responses for each review.
>
> **Q1:What is the additional cost of using the proposed method? Please discuss the efficiency and the other possible cost.**
>
> Re: The extra cost only occurs during training. However, the testing time is exactly the same as that of the baseline model. In practice, training models for short-term predictions is easier and fast to converge. Training long-term predictions in our multi-stage training strategy also converge fast since the model trained for earlier stages are employed as pretrained model to provide prior information. We will include the discussion in the revision.
>
> |                   | **stage 1** | **stage 2** | **stage 3** | **total** |
> | ----------------- | ----------- | ----------- | ----------- | --------- |
> | **Baseline**      | 16h         | -           | -           | 16h       |
> | **Baseline+Ours** | 8h          | 6h          | 5h          | 19h       |
>
> **Q2:Lacking a vivid figure to illustrate the whole method pipeline before the detailed method introduction. Besides, the introduction part, especially the method description can be organized better and more logically.**
>
> Re: Thank you for your suggestion. We will provide a figure to illustrate our pipeline and improve our presentation.
>
> **Q3:A more detailed and clear explanation is essential to illustrate what is knowledge really, or what sign we can get to observe the knowledge utilized or not.**
>
> Re: The knowledge refers to the information involved in the model, which is exploited not only for completing the current prediction task but also enhancing the performance of subsequent tasks. We can assess whether this knowledge has been utilized through the accuracy of long-term predictions. More details can be found in Q5.
>
> **Q4:Fig. 3: hard to read and discover the difference between methods.**
>
> Re: In Fig. 3, the upper image represents the actions of directions, with the person maintaining an upright position throughout the sequence. The results of PGBIG exhibits a bent posture during long-term predictions, whereas our method can predict states closer to the ground truth (GT) position. While in the lower image, the person first bends and then stands upright again. The PGBIG method maintains the bent posture throughout the long-term prediction and our method can accurately predict the changes in posture. It is evident that our method demonstrates a improvement in long-term prediction effectiveness. Thanks for your suggestion and we will highlight the major differences in visualization results among different methods.
>
> **Q5:Lacking direct and clear experiments to show the effectiveness of the continual learning design, e.g., better avoidance of forgetting, better balance of the short and long predictions, etc.**
>
> Re: Following your suggestion, we have conducted experiments to evaluate the evolution of the performances on the previous stages as training continues on the future stages on the Human3.6M dataset. The results are shown in the table below, where lower values indicate more accurate prediction. As shown, introducing prior compensation factor alleviates the performance degradation from stage 1 to stage 3 of Z1 predictions. Specifically, without prior compensation, the prediction error of Z1 increases by 0.83, whereas with prior compensation, it only increases by 0.27. This result suggests that the prior compensation factor can effectively alleviate the forgetting issue. As a result, Z1 can offer more comprehensive priors for Z2 and Z3 predictions, resulting in better prediction performance compared to the training approach without prior compensation.
>
> **without prior compensation:**
>
> |             | **Z1** | **Z2** | **Z3** |
> | ----------- | ------ | ------ | ------ |
> | **Stage 1** | 9.03   | -      | -      |
> | **Stage 2** | 9.44   | 45.33  | -      |
> | **Stage 3** | 9.86   | 45.70  | 92.80  |
>
> **Ours:**
>
> |             | **Z1** | **Z2** | **Z3** |
> | ----------- | ------ | ------ | ------ |
> | **Stage 1** | 9.03   | -      | -      |
> | **Stage 2** | 9.10   | 44.43  | -      |
> | **Stage 3** | 9.30   | 44.62  | 91.37  |
>
> **Q6: typo: Fig. 1: lacking space before the (**
>
> Re: Thank you for pointing out this. We will add space before the missing ( in Fig.1.
>
> **Q7:Though there are some "traditional" settings of the length definition of the long and short terms. I wonder if we change the 5 and 15 frames setting, is the analysis of Fig. 1 kept or changed?**
>
> Re: If we set the short-term to 15 frames, the analysis depicted in Figure 1 still holds. The detailed results are shown in the table. They are consistent with the conclusion drawn from Figure 1, indicating that short-term predictions offer valuable prior information for improving long-term prediction performance.
>
> |            | **short+long** | **short only** | **short then short+long** |
> | ---------- | -------------- | -------------- | ------------------------- |
> | **80ms**   | 10.53          | 9.88           | 10.20                     |
> | **1000ms** | 110.37         | -              | 109.86                    |
>
> **Q8: Learning rate in Eq. 9: choice discussion.**
>
> Re: We have exactly followed the implementation details of the backbone PGBIG and maintained consistency with its learning rate (initialize to 5e-3 and decrease the learning rate exponentially).

---

> > ### Comment · Reviewer_vWPd · 2023-08-14
> > **Post-rebuttal**
> >
> > Thank the authors for the response and additional results. If the next version can be revised according to the promise above, my main concerns are addressed.
> >
> > Looking forward to the other reviewers' discussions.

---

> > > ### Author Response · Authors · 2023-08-16
> > > **Replying to the reviewer's comments**
> > >
> > > Thank you for the response! We highly value your insightful feedback and we will incorporate your suggestions to the subsequent version accordingly.

---

### Official Review · Reviewer_GqjD · 2023-07-01

**Soundness:** 3 good
**Presentation:** 3 good
**Contribution:** 3 good
**Rating:** 5
**Confidence:** 5

**Summary:**

This paper proposes to train human motion prediction networks by gradually increasing the prediction horizon.
This encourages the network to learn short-term predictions first and then leverage the learned to predict longer horizons.
The easy-to-hard curriculum makes the network learn more efficiently, as evidenced by the comparison to networks that learn all horizons at the same time.
Given the continual setting, forgetting prevention is needed and dealt with by the introduced prior compensation factors.
Experiments are carried out on three major benchmarks, and effectiveness is observed compared with selected baselines.

**Strengths:**

The paper is well-written. The idea of training sequence prediction with gradually increased horizons is well instantiated in the context of human motion prediction.
The motivation is also clearly conveyed by the ablation in Figure 1.
Better performance is achieved when compared with two recent methods.
It is good that some derivation is shown to arrive at the final combined loss of predictions at different horizons.
The adaptive scheme from the derivation shows better performance than a hand-crafted fixed set of weights.

**Weaknesses:**

The comparison is a bit weak. Shall compare with more recent methods, for example, "Back to MLP: A Simple Baseline for Human Motion Prediction, 2023."
Also, since the method proposed is not backbone-dependent, so more evaluation is needed, for example, transformer-based architectures.
Moreover, there are some questionable parts in the derivation, even though it seems that the final result may not be heavily affected.

**Questions:**

1) In lines 114-117, all predictions are conditioned on X_1:T_h, however, in Eq. 2, the conditions consist of previous predictions.
2) Eq. 3 assumes that the actual distribution is a degenerated distribution plus a positive bias, why is the bias always positive? This seems not reasonable as the degenerated one could be larger than the actual one.
3) Eq. 4 shows that under the bias assumption, the final loss can be treated as a linear combination of the loss terms for different horizons, however, in Eq. 5 the weights for each term is changed from \alpha to 1-\alpha, is this really valid?
4) Are the terms involving only \alphas really optimized by Eq. 8?
5) Not sure whether the presentation/derivation is really necessary given that a lot of approximation is needed, yet what we want is just a loss that weights the prediction error at different horizons, hopefully the adaptiveness is the key?

**Limitations:**

The limitation is addressed in the broader impact section.

---

> ### Author Rebuttal · Authors · 2023-08-10
>
> **Part 1 (Part 2 is in global rebuttal)**
>
> Thanks to the reviewer for the constructive comments. We have carefully addressed your concerns and provided detailed responses for each review.
>
> **Q1: The comparison is a bit weak. Shall compare with more recent methods, for example, "Back to MLP: A Simple Baseline for Human Motion Prediction, 2023."**
>
> Re: The mentioned approach is a very recent work, and considering your suggestion, we have also conducted experiments with the Human3.6M dataset on this model. The experimental results are shown in the Table 6. It can be observed that our proposed training strategy consistently improves the performances of corresponding models. The average error of the baseline **siMLPe** **[1]** model is 68.76. When training it with our multi-stage strategy, the average error reduces to 67.57. This also verifies that our approach can be flexibly applied to various backbone models for human pose prediction, enhancing their performance for human motion prediction.
>
> **Q2: Since the method proposed is not backbone-dependent, so more evaluation is needed, for example, transformer-based architectures.**
>
> Re: Yes. Our training strategy is not dependent on the backbone and can be flexibly applied to other models. In the manuscript, we have tested it with three different backbones: PGBIG [3] (GCN-based), LTD [4] (GCN-based), and Res. Sup. [5] (LSTM-based). Following your suggestion, we also included a transformer-based backbone and tested it on the Human3.6M dataset. Specifically, we used the POTR [2] as our backbone model, which leverages a transformer as the primary framework for parallel prediction manner. The same as [2], we used Euler Angle Error as the evaluation metric. The experimental results are presented in the Table 7. As expected, our method shows consistent improvements with this backbone as well, further validating that our proposed training strategy is quite flexible and effective to improve different prediction models.
>
> **Q3:There are some questionable parts in the derivation, even though it seems that the final result may not be heavily affected.**
>
> Re: Thank you for the suggestion. We have thoroughly validated our approach both theoretically and experimentally. We will carefully examine the details of the derivations to ensure the accuracy of our conclusions.
>
> **Q4:In lines 114-117, all predictions are conditioned on X_1:T_h, however, in Eq. 2, the conditions consist of previous predictions.**
>
> Re: Actually, the conditions in lines 114-117 and Eq.2 have different meanings. Specifically, in line 114-117, we define the prediction task $Z_k$ as predicting segment $k$, i.e., $X_{T_{Z_{k-1}}+1:T_{Z_k}}$, conditioned on the history $X_{1:T_h}$. While in Eq.2, the probability of $Z_k$ conditioned on $Z_1, …, Z_{k-1}$ suggests exploiting the beneficial knowledge of previous tasks $Z_1, …, Z_{k-1}$ to predict task $Z_k$. We will improve the writing to ensure a more concise and understandable presentation.
>
> **Q5:Eq. 3 assumes that the actual distribution is a degenerated distribution plus a positive bias, why is the bias always positive? This seems not reasonable as the degenerated one could be larger than the actual one.**
>
> Re: Indeed, this bias is always positive. The $P(Z_{k}|Z_{1}Z_{2}\cdots Z_{k-1}\theta)$ represents the most ideal scenario, where the current prediction task can fully leverage the prior information provided by previous prediction tasks. While $P(Z_{k}|\hat{Z}_{1:k-1}\theta)$ indicates completing the current prediction task with incomplete prior information. Hence, utilizing complete predictive priors would yield more accurate predictions than using incomplete information.
>
> **Q6: In Eq. 5 the weights for each term is changed from \alpha to 1-\alpha, is this really valid?**
>
> Re: Yes, it is valid. We have double checked the derivations presented in the appendix material.
>
> **Q7: Are the terms involving only \alphas really optimized by Eq. 8?**
>
> Re: We actually optimize $\alpha$s by leveraging Eq.7 as the loss function. Eq.8 is used to estimate the $\hat{\alpha}_{Z_{1:k-1}\rightarrow Z_{k}}$  involved in Eq. 7 for the training in subsequent stages. The detailed training process can be found in Algorithm 1.
>
> **Q8: Not sure whether the presentation/derivation is really necessary given that a lot of approximation is needed, yet what we want is just a loss that weights the prediction error at different horizons, hopefully the adaptiveness is the key?**
>
> Re: Thank you for your insightful comments. The derivation is necessary. Aiming to compensate for the loss of prior knowledge when switching stages, we introduce α and obtain Eq. 3. Only by relying on Eq.3 can we derive the final objective function with the form of Eq.5. Although the objective function appears to be dynamic weighting control, the underlying mechanism is derived from rigorous theoretical analysis. Specifically, our strategy promote the model training process by estimating the extent of prior information loss. This estimation can only be achieved through a multi-stage training process. In contrast, the assignment of different task weights can be accomplished through a one-stage training approach. Moreover, arbitrarily designed dynamic weight control often fails to improve the training process effectively.
>
> To validate the effectiveness of our derived objective function, we also conducted an experiment using a simple dynamic weighting approach in one-stage training manner. The results demonstrate that our strategy achieves better performance, which is shown in Table 8.

---

> > ### Comment · Reviewer_GqjD · 2023-08-14
> >
> > Thanks for the response. Will keep my current rating and hope the authors can make the modifications as promised in future versions.

---

> > > ### Author Response · Authors · 2023-08-16
> > > **Replying to the reviewer's comments**
> > >
> > > Thank you for the response. We are grateful for your insightful review and we will incorporate your suggestions to the later version accordingly.

---

### Official Review · Reviewer_hcCQ · 2023-07-07

**Soundness:** 2 fair
**Presentation:** 2 fair
**Contribution:** 2 fair
**Rating:** 5
**Confidence:** 3

**Summary:**

This paper aims to enhance human motion prediction. The main contributions of this paper are:
1. The paper presents a multi-stage training strategy named Temporal Continual Learning to incorporate the learning of both short-term prediction and long-term prediction.
2. The paper introduces Prior Compensation Factor to better preserve prior information during the process of Temporal Continual Learning.
These optimizations are given through theoretical derivation.


**Strengths:**

The paper has several strengths:
1. Overall, the paper is well-written with a clear and well-motivated introduction. The storyline to leverage the prior knowledge learned from short-term inputs to facilitate long-term predictions makes sense.
2. Secondly, the proposed method is flexible and demonstrates good performance when applied to different human motion predictors, outperforming state-of-the-art on different datasets.


**Weaknesses:**

The paper could benefit from a more thorough discussion of related work on Human Pose Prediction, such as in L23-24, transformer architectures include
- PoseGPT: Quantization-Based 3D Human Motion Generation and Forecasting. ECCV 2022

	and graph convolution networks include

- Diverse Human Motion Prediction Guided by Multi-Level Spatial-Temporal Anchors. ECCV 2022

The key insight is supported by L32-42 and Figure 1. I'm wondering if the comparison can be represented by a clearer illustration, e.g. using figures to explain different terms such as “short+long” and “short then short + long”. The current version looks a bit rough.


**Questions:**

1. typo in L119, should be T_{Z_{K-1}}
2. Figure 4 in the ablation study is a bit confusing. What is ablated here? Is the alpha value the same across different experiments?
3. The implementation defines a specific partition of sequence for the multi-task. I think it would be interesting to see the ablation on different partitions, e.g. different number of sub-tasks


**Limitations:**

It would be helpful if the authors could provide videos to demonstrate the quality of generated human motion. I wonder if the multi-task learning for different motion segments will cause the discontinuity

---

> ### Author Rebuttal · Authors · 2023-08-10
>
> Thanks to the reviewer for the constructive comments. We have carefully addressed your concerns and provided detailed responses for each review.
>
> **Q1:** **The paper could benefit from a more thorough discussion of related work on Human Pose Prediction, such as in L23-24, transformer architectures.**
>
> Re: Thank you for the suggestion. Some works attempt to develop human motion prediction models based on Transformer architectures. For instance, PoseGPT [1] focus on generating diverse future poses with GPT-like model. [2] introduces spatial-temporal anchor-based sampling to generate diverse human poses. We will include these works and provide a more thorough discussion in the revision.
>
> **Q2:** **The key insight is supported by L32-42 and Figure 1. I'm wondering if the comparison can be represented by a clearer illustration, e.g. using figures to explain different terms such as “short+long” and “short then short + long”. The current version looks a bit rough.**
>
> Re: "short+long" means that the model are trained for both short-term and long-term predictions together. "short only" indicates that the model is only trained for short-term predictions without considering long-term predictions. "short then short+long" means that after training the model for short-term prediction, the model is further trained by combining both short-term and long-term predictions together. We will follow your suggestion and provide a figure to illustrate the differences of these terms.
>
> **Q3:** **typo in L119, should be T_{Z_{K-1}}**
>
> Re: Thank you. We will correct the writing error.
>
> **Q4:** **Figure 4 in the ablation study is a bit confusing. What is ablated here? Is the alpha value the same across different experiments?**
>
> Re: The purpose of the figure is to visualize the average value of α for each stage. As can be seen, as the training stages gets larger, more prior information loss can be observed, and our multi-stage training strategy can effectively reduce the information loss. Since alpha is learned in the data-driven manner, its value could vary in different experiments.
>
> **Q5:** **The implementation defines a specific partition of sequence for the multi-task. I think it would be interesting to see the ablation on different partitions, e.g. different number of sub-tasks.**
>
> Re: Thank you for your suggestion. We have conducted experiments with different numbers of tasks, and the results are shown in the following table. As can be observed, the model's performance improves as the number of tasks gets larger from 1 to 3, and it remains stable when the number of tasks becomes larger than 3.
>
> | **number of tasks** | **1** | **2** | **3** | **5** | **8** |
> | ------------------- | ----- | ----- | ----- | ----- | ----- |
> | **Avg_err⬇**        | 66.95 | 66.02 | 65.00 | 65.05 | 65.03 |
>
> **Q6:** **It would be helpful if the authors could provide videos to demonstrate the quality of generated human motion. I wonder if the multi-task learning for different motion segments will cause the discontinuity.**
>
> Re: In the proposed multi-stage learning strategy, the training process does not change the way of model prediction (parallel or auto-regressive). We only changed the way of determining parameters of the model. As a result, we did not break the continuity of the results generated by the original model. We also examine the generated motion sequence and find that it is visually continuous. And we have provided a link of our demo to AC.
>
> **reference:**
>
> [1] Lucas, T., Baradel, F., Weinzaepfel, P., & Rogez, G. (2022, October). Posegpt: Quantization-based 3d human motion generation and forecasting. In European Conference on Computer Vision (pp. 417-435). Cham: Springer Nature Switzerland.
>
> [2] Xu, S., Wang, Y. X., & Gui, L. Y. (2022, October). Diverse human motion prediction guided by multi-level spatial-temporal anchors. In European Conference on Computer Vision (pp. 251-269). Cham: Springer Nature Switzerland.

---

### Official Review · Reviewer_skai · 2023-07-07

**Soundness:** 3 good
**Presentation:** 2 fair
**Contribution:** 3 good
**Rating:** 6
**Confidence:** 4

**Summary:**

The paper addresses the common trade-off between the short- and long-term prediction quality of 3D human motion prediction models. The proposed training technique improves the performance of the underlying models both on short- and long-term prediction horizons, where the models generally prioritize one and suffer from the other. It is a multi-stage training scheme resembling curriculum learning with an increasing prediction horizon. The prediction horizon is split into consecutive chunks of frames (i.e., stages). The model is trained iteratively for every stage by initializing the weights from the previous stage and considering all the stages so far. Different from curriculum learning, the proposed technique aims to preserve and leverage the prior information of the past stages by alleviating catastrophic forgetting in the future stages. To do so, the paper incorporates the "Prior Compensation Factor" in the training objective. Intuitively, for every training sample, the model predicts a weight parameter that dynamically distributes the total loss weight across stages. The proposed technique is evaluated using three different architectures. Experiments on various benchmarks show that it is highly effective.

**Strengths:**

**originality**
The paper addresses a common and often neglected problem in the 3D human motion prediction domain. In fact, most prior works prefer to ignore the short- and long-term prediction trade-off and present results with separate model configurations. I find this effort valuable.

**quality**
The proposed technique is well-motivated and sound. I have some concerns regarding the theoretical analysis (see below), but the experimental results are solid.

**clarity**
The motivation is clear. The paper provides enough background to understand the problem setting. However, it is not straightforward to intuitively understand what the proposed technique does.

**significance**
I find the paper interesting and useful for the community. The experiments show that the proposed technique could improve the performance of the out-of-the-box models.


**Weaknesses:**

There is a potential problem with lemma 3.1. It expects `b` to be between 0 and 1, a likelihood value from a probability mass function. For the proposed training objective (Eq. 7) to be held, the loss functions should be chosen carefully. In this work, it is implemented as a MSE loss (Eq. 6 and 7), which does not have an upper-bound. How about implementing the objective as a log-likelihood with a Gaussian output model? It may have negative values.

I had to spend some time getting an intuitive understanding of Eq. 7 (no, the “An intuitive explanation” section does not help). It boils down to a dynamic weighting scheme where the network learns to distribute the loss weights across prediction horizons (i.e. stages). I think this could be explained better in the paper. In fact, the training dynamics is very straightforward. If the MSE loss is high, then the predicted alpha should be higher to optimize the objective. We can say that the model learns to “assess the difficulty of the task”. Naturally, alpha values get higher in the future stages as it becomes more difficult. If you run the following code snippet, you get a very similar plot to the one in Fig. 4 ($\alpha$ values at different stages). It gives you the optimal $\alpha$ value for different MSE values. I kindly ask the authors to clarify if I am wrong or share their thoughts on this analysis.

```
import matplotlib.pyplot as plt
from scipy.optimize import minimize_scalar
import math
import numpy as np

def objective(mse):
    return lambda alpha: (1 - alpha) * mse + (1-alpha) * math.log(1-alpha) + math.log(1 + alpha)

mse_values = np.arange(0, .5, 0.01)
best_alphas = []
best_res = []
for mse in mse_values:
    opt_res = minimize_scalar(objective(mse), bounds=(0, 1), method="bounded")
    best_alphas.append(opt_res.x)
    best_res.append(opt_res.fun)

plt.plot(mse_values, best_alphas)
plt.xlabel("MSE values (i.e., stages)")
plt.ylabel("Alpha values")
plt.show()
```
The authors can also try setting the $\alpha$ values for the “HC” ablation using this code snippet.

An ablation on the number of stages is missing. Similarly, its cost is not discussed. While it improves the performance of the underlying model, it introduces a significant overhead at training time.

It would be interesting to see the evolution of the performances on the previous stages as training continues on the future stages. A triangular matrix reporting the performance of stage K after training every stage > K.

Instead of following a dynamic approach (i.e., predicting an $\alpha$ per training sample), would it still work if a single, stage-wise $\alpha$ variable was trained?

This is merely a suggestion for the presentation. I think it would be more interesting to report the performance for the pairs (baseline, baseline + ours) in all benchmarks (as in Table 5).


**Questions:**

I’ve raised my concerns and asked my questions in the weaknesses section.

**Limitations:**

The paper does not have a limitations section. I think it is clear that the proposed technique significantly increases training complexity.

---

> ### Author Rebuttal · Authors · 2023-08-10
>
> Thanks to the reviewer for the constructive comments. We have carefully addressed your concerns and provided detailed responses for each review.
>
> **Q1:** **There is a potential problem with lemma 3.1. It expects b to be between 0 and 1, a likelihood value from a probability mass function. For the proposed training objective (Eq. 7) to be held, the loss functions should be chosen carefully. In this work, it is implemented as a MSE loss (Eq. 6 and 7), which does not have an upper-bound. How about implementing the objective as a log-likelihood with a Gaussian output model? It may have negative values.**
>
> Re: We would like to clarify that in our implementation, the objective is formulated as log-likelihood with a Gaussian output, in which the model output is defined by a Gaussian distribution with value from 0 to 1. Thus, the requirements of lemma 3.1 are explicitly satisfied. Then, we take the negative logarithm of the Gaussian distribution and derive the MSE loss, whose value range between [0, +∞).
>
> **Q2: α** **values get higher in the future stages as it becomes more difficult. It's like the model is learning to evaluate task difficulty. I kindly ask the authors to clarify if I am wrong or share their thoughts on this analysis.**
>
> Re: Our perspective on this differs slightly from yours. Rather than evaluating difficulty of the current task itself, our proposed strategy estimates the extent of prior information loss, which can also be viewed as estimating the difficulty of transferring prior information from previous tasks to the current one. Our motivation is to effectively utilize the prior knowledge acquired from past tasks. To achieve this, we employ a multi-stage training strategy and introduce a prior compensation factor to estimate the extent of information loss.It is worth noting that the multi-stage training method allows us to estimate the degree of prior information loss. This estimation can also be seen as measuring the difficulty of transferring prior knowledge of previous tasks to the current one. In contrast, evaluating task difficulty to assign varying task weights can be achieved in one stage. This is a significant difference.
>
> **Q3:** **The authors can also try setting the α values for the “HC” ablation using this code snippet.**
>
> Re: Thank you for your suggestion. We have conducted an experiment, referred to as "HC2", using the α values calculated with the code you provided to validate this viewpoint. The results are shown in Table 1, from which we can obtain two observations. First, the analytical α calculated by “HC2” also helps to mitigate the loss of prior information. Second, our learning strategy performs better than “HC2”. It is because that in our model learning process, α is treated as a distribution that estimates the loss of prior information for previous stages. Based on the α definition, given that the model's output is inherently a distribution, the α should be considered as a random variable satisfying a certain distribution as well. Therefore, using the model to estimate the distribution of α is a more reasonable way.
>
> **Q4: An ablation on the number of stages is missing.**
>
> Re: Following your suggestion, we have conducted experiments with different numbers of stages, and the results are presented in Table 2. As can be seen, the model's performance improves as the number of stages gets larger from 1 to 3, and it remains stable when the number of stages becomes larger than 3.
>
> **Q5: It would be interesting to see the evolution of the performances on the previous stages as training continues on the future stages. A triangular matrix reporting the performance of stage K after training every stage > K.**
>
> Re: Based on your suggestion, we conducted experiments on the Human3.6M dataset, as shown in the Table 3 (lower values indicate smaller errors). It can be observed that as the training continues, the performances of previous prediction tasks slightly decrease. Taking Z1 task for example, the performances decrease from 9.03 to 9.30.
>
> **Q6: Instead of following a dynamic approach (i.e., predicting an α per training sample), would it still work if a single, stage-wise α variable was trained?**
>
> Re: We conduct experiments based on your suggestion, using the code you provided to calculate the average α value as stage-wise α. The results are shown in the Table 4. It shows that the stage-wise α helps in mitigating the loss of prior information and aiding the model's training process. However, this approach isn't as effective as our strategy, largely because the model approximates the loss of prior information at each stage in a coarse-grained manner, leading to imprecise control over the training process.
>
> **Q7: I think it would be more interesting to report the performance for the pairs (baseline, baseline + ours) in all benchmarks (as in Table 5).**
>
> Re: Thank you for your suggestion. We will report the performances in the form of **baseline and baseline+ours** **pairs**.
>
> **Q8：Its cost** **is not discussed. While it improves the performance of the underlying model, it introduces a significant overhead at training time.**
>
> Re: Thank you for your comments. We would like to point out that our multi-stage training approach does not lead to a significant increase in the overall training time as compared with the single stage training approaches (16h vs. 19h). The detailed training time on Human3.6M dataset can be found in the Table 5. In practice, training models for short-term predictions is easier and fast to converge. Training long-term predictions in our multi-stage training strategy also converge fast since the model trained for earlier stages are employed as pretrain model to provide prior information. We will include the discussion in the revision.

---

> > ### Comment · Reviewer_skai · 2023-08-15
> > **Post-rebuttal comment**
> >
> > I thank the authors for their rebuttal and additional results! Their rebuttal provides clarifications and addresses my main concerns.
> >
> > I think interpretation of the $\alpha$ term depends on the perspective. I see the point in the authors' interpretation. I do not have further questions. I will follow the discussion with other reviewers and revise my score accordingly.

---

> > > ### Author Response · Authors · 2023-08-16
> > > **Replying to the reviewer's comments**
> > >
> > > Thank you for the response. We appreciate your valuable review and we will incorporate your suggestions to the later version accordingly.

---

### Author Rebuttal · Authors · 2023-08-10

**(Part 2 for reviewer GqjD)**

**reference:**

[1] Guo, W., Du, Y., Shen, X., Lepetit, V., Alameda-Pineda, X., & Moreno-Noguer, F. (2023). Back to mlp: A simple baseline for human motion prediction. In Proceedings of the IEEE/CVF Winter Conference on Applications of Computer Vision (pp. 4809-4819).

[2] Martínez-González, A., Villamizar, M., & Odobez, J. M. (2021). Pose transformers (potr): Human motion prediction with non-autoregressive transformers. In Proceedings of the IEEE/CVF International Conference on Computer Vision (pp. 2276-2284).

[3] Ma, T., Nie, Y., Long, C., Zhang, Q., & Li, G. (2022). Progressively generating better initial guesses towards next stages for high-quality human motion prediction. In Proceedings of the IEEE/CVF Conference on Computer Vision and Pattern Recognition (pp. 6437-6446).

[4] Mao, W., Liu, M., Salzmann, M., & Li, H. (2019). Learning trajectory dependencies for human motion prediction. In Proceedings of the IEEE/CVF international conference on computer vision (pp. 9489-9497).

[5] Martinez, J., Black, M. J., & Romero, J. (2017). On human motion prediction using recurrent neural networks. In Proceedings of the IEEE conference on computer vision and pattern recognition (pp. 2891-2900).


**(Part 2 for reviewer aXof)**

**Q9: Efficiency and multi-stage pipeline. Recent researches [5] suggest predicting motions in one stage, which is easier to train. Will the multi-stage training be harder to tune or train? Will it be more time-consuming? I would like to discuss with the authors about it.**

Re: The multi-stage training approach we proposed does not require complex adjustment techniques. What's more, it offers certain advantages over single-stage training. The temporal multi-stage training process leverages prior predictive information from earlier stages more effectively, thereby guiding the learning of challenging long-term prediction. Simultaneously, training the short-term prediction separately is easier to learn and results in better short-term predicting capabilities. Although our proposed strategy incurs increase in training time, the testing time remains consistent with that of the backbone model. Regarding the training process, due to the faster convergence of short-term predictions, fewer iterations are needed for training the first stage. Additionally, because the model can better utilize the prior information provided by short-term predictions, the subsequent stages of training also converge more easily. As a result, the overall training time does not increase significantly. The training time of different stages are demonstrated in the Table 5. We will include this discussion in the subsequent manuscript.

**Q10: It will be great if authors can discuss or provide zero-shot adaptation experiments on other datasets.**

Re: We conducted zero-shot experiments based on your advice. Due to variations in the number of key-points and annotations across different human motion datasets, we conduct the zero-shot experiments on Human3.6M dataset with backbone only trained on stage 1 (task Z1). The results can be found in the table below. First, the zero-shot performance is not as good as training all tasks. Second, there are variations in performance between different backbones, with PGBIG outperforming others.

|                                | **Z2** | **Z3** |
| :----------------------------: | :----: | :----: |
|       **Res. Sup.+Ours**       | 83.23  | 146.80 |
| **Res. Sup.+Ours (zero-shot)** | 86.26  | 155.66 |
|         **PGBIG+Ours**         | 44.39  | 91.11  |
|   **PGBIG+Ours (zero-shot)**   | 76.84  | 122.59 |

**reference:**

[1] Zhong, C., Hu, L., Zhang, Z., Ye, Y., & Xia, S. (2022). Spatio-temporal gating-adjacency gcn for human motion prediction. In Proceedings of the IEEE/CVF Conference on Computer Vision and Pattern Recognition (pp. 6447-6456).

[2] Sofianos, T., Sampieri, A., Franco, L., & Galasso, F. (2021). Space-time-separable graph convolutional network for pose forecasting. In Proceedings of the IEEE/CVF International Conference on Computer Vision (pp. 11209-11218).

[3] Bouazizi, A., Holzbock, A., Kressel, U., Dietmayer, K., & Belagiannis, V. (2022). Motionmixer: Mlp-based 3d human body pose forecasting. arXiv preprint arXiv:2207.00499.

[4] Guo, W., Du, Y., Shen, X., Lepetit, V., Alameda-Pineda, X., & Moreno-Noguer, F. (2023). Back to mlp: A simple baseline for human motion prediction. In Proceedings of the IEEE/CVF Winter Conference on Applications of Computer Vision (pp. 4809-4819).

[5] Chen, L. H., Zhang, J., Li, Y., Pang, Y., Xia, X., & Liu, T. (2023). HumanMAC: Masked Motion Completion for Human Motion Prediction. arXiv preprint arXiv:2302.03665.

[6] Martínez-González, A., Villamizar, M., & Odobez, J. M. (2021). Pose transformers (potr): Human motion prediction with non-autoregressive transformers. In Proceedings of the IEEE/CVF International Conference on Computer Vision (pp. 2276-2284).

[7] Ma, T., Nie, Y., Long, C., Zhang, Q., & Li, G. (2022). Progressively generating better initial guesses towards next stages for high-quality human motion prediction. In Proceedings of the IEEE/CVF Conference on Computer Vision and Pattern Recognition (pp. 6437-6446).

[8] Martinez, J., Black, M. J., & Romero, J. (2017). On human motion prediction using recurrent neural networks. In Proceedings of the IEEE conference on computer vision and pattern recognition (pp. 2891-2900).

---

### Decision · Program_Chairs · 2023-09-21

**Decision:**

Accept (poster)

**Comment:**

The manuscript proposes a multi-stage training scheme resembling curriculum learning with an increasing prediction horizon for human motion prediction.

The reviewers acknowledge that the paper addresses a common but often neglected problem in human motion prediction, that the proposed technique is well-motivated and sound, and that the proposed technique improves the performance of various models.

The reviewers had some questions regarding the interpretation of the $\alpha$ term and Figure 1. They suggested a more thorough discussion of related work and requested to include more recent methods and datasets in the comparison.

During the discussion, the authors answered the questions of the reviewers and resolved their concerns. While there was some confusion regarding the experiments on the AMASS dataset and the submitted results in the PDF of the rebuttal due to missing captions, the additional results reported during the discussion support the results reported in the paper and show the usefulness of the proposed training scheme, which can be applied to a large range of approaches for human motion prediction. The authors are asked to include the additional results in the camera-ready version and address the clarity issues raised by the reviewers.